# Self-cleaving guide RNAs enable pharmacological selection of precise gene editing events in vivo

Amita Tiyaboonchai [1,2] ✉, Anne Vonada[1,3], Jeffrey Posey [1], Carl Pelz[2], Leslie Wakefield [1] & Markus Grompe [1,2,3]

Expression of guide RNAs in the CRISPR/Cas9 system typically requires the use of RNA polymerase III promoters, which are not cell-type specific. Flanking the gRNA with self-cleaving ribozyme motifs to create a self-cleaving gRNA overcomes this limitation. Here, we use self-cleaving gRNAs to create drug-selectable gene editing events in specific hepatocyte loci. A recombinant Adeno Associated Virus vector targeting the Albumin locus with a promoterless self-cleaving gRNA to create drug resistance is linked in cis with the therapeutic transgene. Gene expression of both are dependent on homologous recombination into the target locus. In vivo drug selection for the precisely edited hepatocytes allows >30-fold expansion of gene-edited cells and results in therapeutic levels of a human *Factor 9* transgene. Importantly, self-cleaving gRNA expression is also achieved after targeting weak hepatocyte genes. We conclude that self-cleaving gRNAs are a powerful system to enable cell-type specific in vivo drug resistance for therapeutic gene editing applications.

Precise gene editing by recombinant adeno-associated virus (rAAV) is an attractive approach for gene therapy[1,2]. Promoterless GeneRide (GR) rAAV vectors create a targeted integration at a designated locus by homologous recombination[3]. The therapeutic transgene is driven by the endogenous genomic promoter only after proper recombination resulting in cell-type specific expression and avoiding the potential activation of oncogenes after random integration. However, the a priori efficiency of GR rAAV vectors is fairly low («1%) and results in only a small population of hepatocytes acquiring the desired gene edit[4]. Our lab therefore previously developed a GR vector linking a desired genetic modification to a drug-selectable gene disruption cassette in cis[5,6]. Using this system, we could select and expand hepatocytes with the required gene edit from subtherapeutic to fully therapeutic levels[6].

Our previous method utilized a short-hairpin RNA (shRNA)-mediated gene knockdown to enable drug selection. The protective shRNA that rendered cells drug resistant was included in the GR vector in cis downstream of the therapeutic cDNA. However, shRNAs are typically driven by RNA polymerase III (RNAP III) promoters, most commonly U6[7] and H1[8]. In order to maintain the promoterless nature of the GR vector, the selectable gene disruption cassette must also be expressed from the endogenous polymerase II (RNAP II) promoter. Transcription by RNAP II is accompanied by post-transcriptional modifications including 5′ capping, polyadenylation, and chemical modifications to the RNA product[9]. These modifications affect the precise sequence of the shRNA and its functionality[7]. To enable proper processing of the shRNA after transcription by RNAPII, it was embedded within a microRNA (miR) scaffold[10,11]. After recombination of the GR into the target *Albumin* locus, this resulted in efficient knockdown of the drug-selectable target gene, thereby enabling efficient selection of the gene-edited hepatocytes[6]. While this approach was effective when targeting the highly expressed *Albumin* gene, it is unlikely that

[1]Oregon Stem Cell Center, Papé Pediatric Research Institute, Oregon Health & Science University, Portland, OR 97239, USA. [2]Department of Pediatrics, Oregon Health & Science University, Portland, OR 97239, USA. [3]Department of Molecular and Medical Genetics, Oregon Health & Science University, Portland, OR 97239, USA. ✉e-mail: tiyaboon@ohsu.edu

weaker genes can drive sufficiently high levels of shRNA to achieve selectable knockdown. Gene editing of a locus with a weak RNAP II promoter would likely achieve only partial knockdown of gene expression using this system[12]. Therefore, we sought to develop a system that would enable drug selection of hepatocytes with precise gene editing event in any hepatocyte locus of medical interest.

In contrast to shRNAs, the CRISPR/spCas9 system utilizing gene-specific guide RNAs (gRNAs) creates insertion/deletion mutations (indels) that result in complete loss of function of the targeted gene without the necessity for a strong promoter. Additionally, design of efficient gRNAs has proved much more amenable to computational modeling, and multiple algorithms exist to predict their functional efficiency[13–15]. Similar to shRNAs, gRNA expression is typically driven by RNAP III promoters to produce a precise RNA sequence without post-transcriptional modifications. This limitation makes it difficult to use the CRISPR/spCas9 system in a cell type-specific manner. It is now possible to express a gRNA from a RNAP II promoter by using self-cleaving guide RNAs (scgRNA)[16]. The scgRNA consists of a gRNA flanked by two self-cleaving ribozymes: a hammerhead (HH) ribozyme located 5′ to the gRNA and a hepatitis delta virus (HDV) ribozyme located 3′ to the gRNA. Ribozymes are sequences of RNA that, once transcribed and folded into their secondary structure, have autocatalytic activity causing cleavage of the RNA sequence at a specific site[17]. By embedding a single gRNA between these two ribozymes, the precise cleavage following transcription allows the release of a fully functional gRNA. These ribozymes have previously been shown to function in vitro in the HEK293 cell line and in vivo in *Leishmania donovani*, mice, yeast, zebrafish, and rice to effectively generate a functional gRNA from a RNAP II promoter[16,18–24].

Here we demonstrate in vivo application of RNAP II-driven scgRNAs for gene therapy by precise homologous recombination. We show that expansion of hepatocytes containing an *Albumin* (*Alb*) targeted integrating recombinant adeno-associated virus (rAAV) vector with the therapeutic gene human Factor 9 and a selectable scgRNA results in highly efficient selection of hepatocytes containing the correctly targeted vector. In order to explore the dynamic range of the functional scgRNA expression from an endogenous promoter, expression of the scgRNA from hepatocyte-specific genes that are more weakly expressed than *Alb*, including *Transthyretin* (*Ttr*), *Phenylalanine hydroxylase* (*Pah*), and *Factor 9* (*F9*), were also tested. In these lower expressing loci, the scgRNA was also expressed and functional. Together these data demonstrate the powerful ways in which scgRNAs can be utilized to allow the specific expression of a CRISPR/spCas9 gRNA from a RNAP II promoter in vivo.

## Results

### Selection of an albumin-targeting gene ride vector in FAH mice
GR vectors are an rAAV containing a therapeutic gene flanked by arms of homology that target vector integration to a specific genomic locus. GR vectors are promoterless and rely upon endogenous promoters for expression of the therapeutic gene following precise integration[3]. For initial application of the GR-scgRNA system, we utilized a Fumarylacetoacetate hydrolase deficient (*Fah*⁻/⁻) mouse model. This mouse shows selective expansion of corrected hepatocytes due to cellular injury in the *Fah*⁻/⁻ hepatocytes. Fah is an enzyme in the tyrosine catabolic pathway involved in the degradation of Fumarylacetoacetate (FAA). The loss of Fah results in the hepatotoxic accumulation of FAA. To prevent hepatoxicity, mice may be maintained on the drug 2-[2-nitro-4(trifluoromethyl) benzoyl]cyclohexane-1,3-dione (NTBC), which inhibits the upstream enzyme 4-OH-phenylpyruvate dioxygenase (Hpd) and prevents FAA accumulation[25] (Fig. 1a). It has been previously demonstrated that a selectable cassette in the form of an miR-embedded shRNA[6] against *Hpd* can be included such that the rAAV GR vector can be initially recombined into a small population of

hepatocytes in *Fah*⁻/⁻ mice and expand with selective pressure after withdrawal of NTBC.

To determine if a scgRNA is sufficient for selection of precisely integrated GR vectors, neonatal *Fah*⁻/⁻ mice were injected with a GR-*Hpd* scgRNA rAAV consisting of arms of homology to the *Alb* locus flanking a P2A ribosomal skipping sequence, a codon optimized human *F9* (*hF9*) transgene, and a scgRNA targeting *Hpd* (Fig. 1b). At 3 weeks of age, rAAV expressing *Streptococcus pyogenes* Cas9 (spCas9) was delivered into experimental mice (*n* = 7). Withdrawal from NTBC treatment was started two weeks later to select for hepatocytes with correctly integrated GR vectors. At baseline, blood hF9 concentration was similar (*p* = 0.370) between the groups that did and did not receive spCas9 (48 ± 35 ng/mL and 29 ± 20 ng/mL, respectively). Throughout NTBC withdrawal, there was an increase in hF9 levels in the group that received spCas9, and at terminal harvest hF9 levels reached supra-physiological levels of 16,928 ± 9615 ng/mL (Fig. 1c). This was significantly higher (*p* = 0.008) than in the mice that did not receive spCas9 where hF9 levels remained close to baseline levels at 50 ± 54 ng/mL.

At terminal harvest, immunohistochemistry revealed healthy hepatocytes (Fig. 1d). Indels in *Hpd* were analyzed by the Tracking of Indels by Decomposition (TIDE) algorithm[26]. Mice that had received spCas9 had an average of 73 ± 19% indel frequency compared to 1.1 ± 0.5% in the no spCas9 group (*p* = 0.0005) (Fig. 1e). Expression of *Hpd* mRNA in mice that had received spCas9 was approximately 38 ± 12% (*p* = 0.018) of wildtype levels by qPCR (Fig. 1f). Immunofluorescent staining for hF9 confirmed these results, showing large clonally expanded hF9 positive regions covering on average 74.9 ± 8.0% of the liver sections in the mice that had received spCas9 (Fig. 1g and Supplementary Table 1). Precise integration of the rAAV vector by homologous recombination was confirmed by PCR (Fig. S1a). The PCR product corresponding to the integrated vector was low in abundance in mice that did not receive spCas9 due to the low integration frequency of the GR-*Hpd* scgRNA in the absence of selective expansion. These experiments demonstrate that delivery of rAAV GeneRide-*Hpd* scgRNA vector in conjunction with rAAV-spCas9 allows selective expansion of a small starting population of hepatocytes in *Fah*⁻/⁻ mice.

### Acetaminophen selection of a Cypor scgRNA
As a clinically applicable method of in vivo hepatocyte selection, we recently developed a strategy that utilizes the common fever medicine acetaminophen (APAP)[5]. Cytochrome P450 reductase (Cypor) is an obligate cofactor of most cytochrome P450 (Cyp) enzymes and is required for the metabolism of APAP[27] to the hepatotoxic intermediate metabolite N-acteyl-p-benzoquinone imine (NAPQI)[28], causing hepatotoxicity at high doses. In the absence of Cypor, APAP cannot be converted to NAPQI and thus a knockout of *Cypor* results in APAP resistance, allowing selective expansion of Cypor-deficient hepatocytes by repeated APAP treatment.

To test this approach, the *Hpd* scgRNA was replaced by a scgRNA targeting *Cypor* in the Albumin-targeted GR. Neonatal wildtype mice received the GR-*Cypor* scgRNA, followed by rAAV-spCas9 of a different serotype at weaning. Selection for correctly targeted hepatocytes by biweekly APAP treatment was started 2 weeks later (Fig. 2a). At baseline, hF9 levels were comparable between the cohort of mice which would undergo APAP selection and the unselected cohort (*p* = 0.344). hF9 levels significantly increased in the selection cohort during the APAP administration regimen. At terminal harvest hF9 levels in the selection group were 7607 ng/mL ± 3828 ng/mL, significantly higher than 14.4 ± 7.63 ng/mL in the unselected group (*p* = 0.006) (Fig. 2b). APAP treatment in mice that received the GR-*Cypor* scgRNA without spCas9 or received no vector did not result in any increase in hF9 concentrations and due to the toxicity of APAP, most of these mice had to be euthanized at earlier timepoints (Fig. S2). *Cypor* indel frequency in the selection group was 12.3 ± 1.71% compared to 0.38 ± 0.24% in the

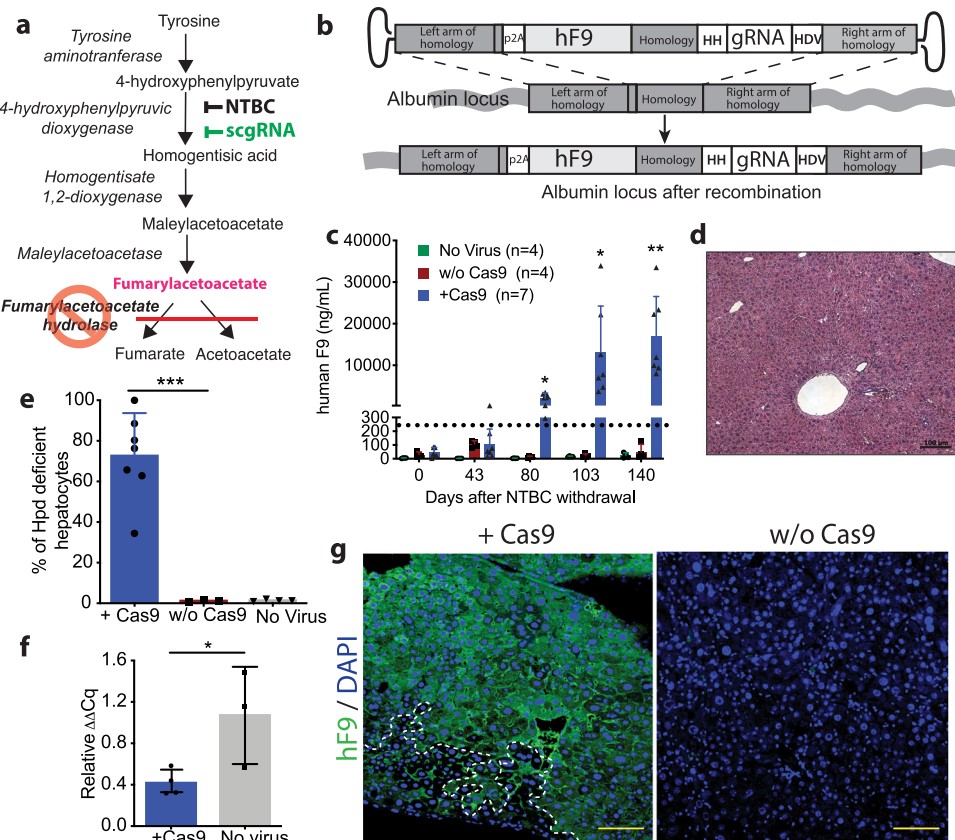

**Fig. 1 | Use of the scgRNA for selection of GR-targeted hepatocytes. a** The tyrosine catabolic pathway. *Fah⁻/⁻* mice lack the ability to metabolize fumarylacetoacetate, resulting in hepatotoxicity. By inhibiting the upstream enzyme 4-hydroxyphenylpyruvate dioxygenase (Hpd) with either the drug NTBC or a scgRNA against *Hpd*, fumarylacetoacetate is not produced. **b** Schematic of the GR-*Hpd* scgRNA rAAV vector and its integration into the Albumin locus by homologous recombination. The rAAV vector contains arms of homology to the Albumin locus, a P2A, and a scgRNA (HH-gRNA-HDV). hF9: human Factor 9; HH: hammerhead ribozyme; gRNA: guide RNA targeting *Hpd*; HDV: hepatitis delta virus ribozyme. Day 80 *P* = 0.012; Day 103 *P* = 0.046; Day 140 *P* = 0.008 (**c**) hF9 concentrations in *Fah-/-* mice treated with ~1 × 10¹⁴ vg/kg GR-*Hpd* scgRNA rAAV that did or did not receive ~8 × 10¹² vg/kg rAAV-spCas9 following NTBC withdrawal. Dashed line indicates therapeutic F9 threshold. Biologically independent mice were analyzed. Data are presented as mean values and error bars represent standard deviation.

**d** Hematoxylin and eosin staining of a representative liver section from a mouse that received spCas9. Scale bar represents 100 μm. *n* = 6 mice. **e** Quantification of *Hpd* deficient hepatocytes by indel frequency. Biologically independent mice were analyzed. Data are presented as mean values and error bars represent standard deviation (*p* = 0.0005). +Cas9 *n* = 7 animals; w/o Cas9 *n* = 3 animals, No virus *n* = 4 animals. **f** Quantification of *Hpd* gene expression levels in the selected mice that received rAAV-spCas9 compared to an untreated wildtype mouse. Biologically independent mice were analyzed. Data are presented as mean values and error bars represent standard deviation *p* = 0.018. +Cas9 *n* = 4 animals; No virus *n* = 3 animals. **g** Immunofluorescent staining for hF9. The dotted line indicates a residual hF9-negative region. Scale bars represent 200 μm. Statistics were calculated using a two-sided unpaired t test. **P* < 0.05, ***P* < 0.01 and ****P* < 0.001. Source data are provided as a Source Data file.

unselected group (*p* = 0.0005) (Fig. 2c). These results were further corroborated by hF9 and Cypor immunofluorescent staining in liver sections from the selected animals that show clonally expanded regions of Cypor-negative, hF9-positive hepatocytes, which are not observed in unselected animals (Fig. 2d). In the selected animals, hF9 staining on average covered 17.2 ± 3.9% of the liver sections (Supplementary Table 1) consistent with the indel frequency from TIDE analysis. Immunohistochemistry revealed healthy hepatocytes with mild accumulation of lipids (Fig. 2e) and Picro Sirius red staining showed rare events of fibrosis in liver sections from the APAP selected group (Fig. 2f). Integration of the targeted rAAV vector into the correct locus was confirmed by PCR (Fig. S1b). These results demonstrate that the GR-*Cypor* scgRNA rAAV can be utilized in conjunction with APAP for selective expansion of precisely gene edited hepatocytes in vivo.

## Processing of GR-Cypor scgRNA

To gain an understanding of the efficiency of cleavage of the ribozymes in the scgRNA cassette, the integrated GR-*Cypor* scgRNA transcript driven from the endogenous Albumin promoter was examined. Initially, a series of northern blots to inspect the *Alb* (Fig. 3a) and *hF9*

(Fig. 3b) transcripts were performed with mouse *alpha-1 antitrypsin* transcript (*mAat*) as a loading control (Fig. 3c). These northern blots demonstrated the presence of the native *Alb* transcript (expected size: 2034 base pairs) as well as the larger molecular weight (MW) fusion transcript of *Alb* and *hF9* from the targeted locus (expected size: 3,415 base pairs). However, the larger MW *hF9* signal required a 6-day exposure to be detectably visualized, compared to 5 min for the endogenous *Alb* signal. The much longer exposure time needed suggested that the fusion transcript was present at much lower abundance than the native *Alb* transcript. As indel analysis indicated that approximately 12.3% of the hepatocytes contained the targeted integration, a stronger signal was expected.

In order to quantitatively compare the relative abundance of transcripts from the targeted and native *Alb* locus, RNA sequencing was performed on GR-*Cypor* scgRNA injected mice with and without APAP selection. In agreement with the observed increase in serum hF9 levels, there was an over 29-fold increase in *hF9* and *P2A* transcripts in APAP selected mice (*n* = 3) compared to unselected mice (*n* = 3) (Fig. 3d). Only a small number of *hammerhead ribozyme, hepatitis delta virus ribozyme* or *spCas9* scaffold transcripts were found, suggesting a

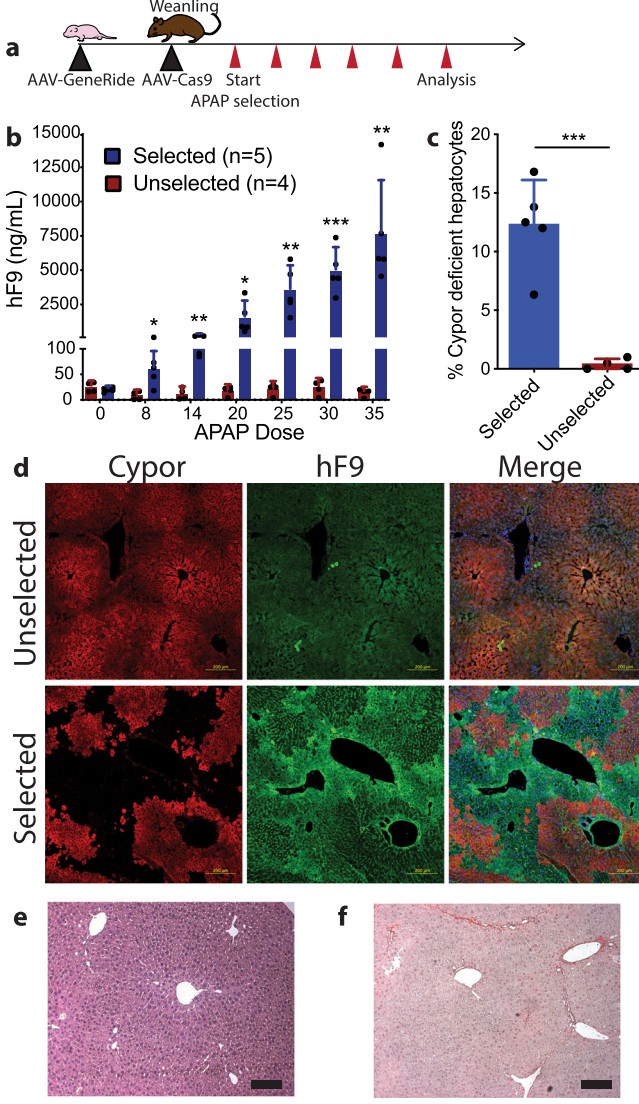

**Fig. 2 | Expansion of hepatocytes with integration of GR-Cypor scgRNA by acetaminophen. a** Schematic of the experimental timeline. The GR-*Cypor* scgRNA rAAV vector containing a hF9 transgene and *Cypor* scgRNA with arms of homology to the Albumin locus was delivered to neonatal WT mice. At weaning, the mice were given rAAV-spCas9 and biweekly APAP administration was started. Mice were bled for hF9 concentration following every 5th dose of APAP administered. **b** hF9 concentrations in mice that were selected with APAP compared to unselected mice. Biologically independent mice were analyzed. Data are presented as mean values and error bars represent standard deviation (Dose 8 $p = 0.015$; Dose14 $p = 0.008$; Dose 20 $p = 0.039$; Dose 25 $p = 0.0051$; Dose 30 $p = 0.0006$; Dose 35 $p = 0.006$). **c** Quantification of Cypor deficient hepatocytes by indel frequency following terminal harvest. Biologically independent mice were analyzed. Data are presented as mean values and error bars represent standard deviation ($p = 0.0005$). Selected $n = 5$ animals; No selection $n = 4$ animals. **d** Representative immunofluorescent staining against Cypor and hF9 in the liver of unselected mice (no APAP treatment) and selected mice (received APAP treatment) at terminal harvest. Scale bars represent 200 μm. **e** Hematoxylin and eosin staining of a liver section from an APAP selected mouse. Scale bars represent 100 μm. $n = 5$ mice. **f** Picro Sirius red staining in an APAP selected mouse. The presence of red staining fibers would indicate fibrosis within the liver. Scale bars represent 100 μm. $n = 5$ mice. Statistics were calculated using a two-sided unpaired t test. *$P < 0.05$, **$P < 0.01$ and ***$P < 0.001$. Source data are provided as a Source Data file.

rapid degradation of these products. These data suggest that while the *Cypor* scgRNA is being transcribed from the targeted loci sufficiently to allow nuclease activity and selection, its presence within the hepatocyte may be highly transient. No differences were observed in *Alb*

mRNA levels in the selected and unselected mice. This was verified by western blot, which showed no significant differences at the protein level of Albumin in APAP selected and unselected mice (Figs. S3a, S3b). Additional genes, including *Cypor*, hepatocyte-specific genes *Ttr* and *Aat*, and housekeeping genes *Gapdh* and *Tbp* also showed no change between selected and unselected samples by RNA sequencing (Fig. S3c).

Quantification of the RNA sequencing data demonstrated that the *hF9* fusion transcript was approximately 11-fold less abundant than native *Alb* in selected mice. This difference corresponds to the weaker *Alb-hF9* band observed in the northern blot of compared to *Alb* mRNA (Fig. 3d). While the total abundance of *hF9* relative to *β-actin*, was $0.732 \pm 0.207$ (Fig. 3d), when normalized to the percentage of Cypor-negative hepatocytes, the *hF9* to *β-actin* ratio was $5.97 \pm 1.33$. The relatively low abundance of hF9 mRNA raised concerns that the scgRNA may be interfering with the expression of the hF9 transcript. As a comparison, a cohort of mice that had received the original selectable GR consisting of a miR embedded shRNA for selection[6] was also analyzed and expression of hF9 was $5.22 \pm 1.48$ ($n = 3$) relative to *β-actin* after normalizing to hepatocytes with GR integrations (Fig. S3d). Hence, the relative expression of *hF9* was comparable in both the mice that had received the scgRNA and the miR-embedded shRNA. This finding indicates that the use of a scgRNA did not reduce the expression of the fusion mRNA relative to a miR-embedded shRNA.

## scgRNA functionality in weakly expressing genes

Albumin is the most highly expressed gene in hepatocytes[29] and this likely explains the supraphysiologic expression levels of hF9 observed from the GR-*Hpd* scgRNA integrated into the *Alb* locus. To test whether functional scgRNA could also be expressed from weaker hepatocyte promoters, GR vectors with arms of homology targeting integration to other loci were tested. The *transthyretin* (*Ttr*), *phenylalanine hydroxylase* (*Pah*), and *Factor 9* (*F9*) genes were chosen, which are 10, 80, and 400-fold lower respectively in expression level relative to *Alb* (Fig. 4a). A control vector lacking arms of homology to the mouse genome was also constructed. In order to sensitively analyze gRNA functionality, the Ai9-tdTomato reporter mouse model was utilized. This mouse contains a transgene in the *Rosa26* locus consisting of a CAG promoter followed by a repetitive stop cassette and tdTomato (Fig. 4b). tdTomato is only expressed following the loss of the stop cassette[30]. A previous study[31] demonstrated that targeting the repetitive region of the stop cassette with a gRNA can cause the CRISPR/spCas9 mediated loss of the stop cassette and subsequent tdTomato expression. GR vectors containing a scgRNA targeting this stop cassette (GR-stop scgRNA) and arms of homology to the 3' UTR of the target genes were generated (Fig. 4b). Unlike in the drug selection experiments described above, homologous recombination into the target locus of and expression of a functional gRNA will result in tdTomato reporter activation, but not APAP resistance.

The various GR-stop scgRNA rAAV vectors were delivered into neonatal Ai9-tdTomato mice that constitutively express spCas9[32]. Mice were harvested at 8 to 12 weeks of age and liver sections were imaged to identify tdTomato-positive hepatocytes. All of the GR-stop scgRNA vectors allowed expression of tdTomato. Small clusters of clonally expanded hepatocytes were present (Fig. 4c). A low frequency of tdTomato positive hepatocytes was also observed in the control rAAV vector with no arms of homology to murine genomic DNA, suggesting baseline leakiness of scgRNA expression. To confirm these findings, hepatocytes were isolated and tdTomato positive cells were quantified by flow cytometry (Fig. 4d and Fig. S4). The percentage of tdTomato positive hepatocytes was significantly higher for *Alb*, *Ttr* and *Pah* targeted rAAV GR vectors when compared to the vector with no homology (Fig. 4g). The *F9*-targeting rAAV GR remained inconclusive.

These experiments were repeated in 3- to 10-week-old adult Ai9-tdTomato mice. Mice were harvested 2–3 weeks following the delivery

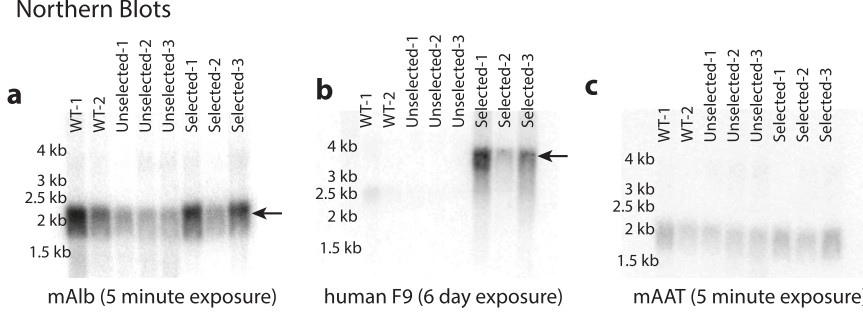

**d**  Next Generation RNA sequencing

| Location | Relative to β-actin | | Ratio of selected/unselected |
| --- | --- | --- | --- |
| | Unselected | Selected | |
| **Alb** | 12.31 ± 5.24 | 8.08 ± 1.44 | 0.66 |
| **Human F9** | 0.025 ± 0.019 | 0.732 ± 0.207 | 29.0 |
| **P2A** | 0.0016 ± 0.0006 | 0.046 ± 0.011 | 29.4 |

**Fig. 3 | RNA analysis of GR-scgRNA treated mice. a–c** Northern blotting of liver RNA from untreated wildtype (WT) mice (n = 2) or mice that recieved GR-*Cypor* scgRNA rAAV with (n = 3) or without (n = 3) APAP selection. **a** Albumin (mAlb), **b** human F9, and **c** Alpha-anti-trypsin (mAat). Each lane represents liver homogenate from one mouse. **d** Table of gene expression from next generation RNA sequencing of mouse Albumin, human F9 or P2A relative to mouse *β-actin* in mice that received the GR-Cypor scgRNA and were APAP selected (n = 3) or remained unselected (n = 3). Error is standard deviation. kb: kilobases; P < 0.05, **P < 0.01 and n.s not significant.

of the rAAV vector. In contrast to neonatal delivery, tdTomato positive hepatocytes were predominantly single cells scattered throughout the liver tissue (Fig. 4e). This was expected as mature hepatocytes rarely undergo cell division without injury[33]. Quantification of tdTomato positive hepatocytes by flow cytometry was consistent with the observations made in neonatally treated mice (Fig. 4f). Targeting the *Alb*, *Ttr* and *Pah* genes resulted in significantly more tdTomato positive hepatocytes compared to the control vector (Fig. 4h). Precise editing of the target loci was confirmed in both neonatal and adult mice by PCR amplification of homology junctions (Fig. S5 and Table 1). We conclude that functional scgRNA resulting in site-specific nuclease activity can be produced from genes with expression levels much lower than Albumin. To also test drug selection from a weaker promoter, a GR-*Cypor* scgRNA targeting *Ttr* was generated. APAP treatment resulted in successful expansion of the population of *Ttr*-targeted hepatocytes (Fig. S6).

## Discussion

While scgRNAs have previously been reported by others[16,19,21–24,34–37], this system has not been exploited for any gene therapy or medical application to date. The CRISPR/spCas9 system is quite efficient in creating gene knockouts and several reports of hepatocyte-specific targeted gene knockouts using this method exist[38,39]. However, therapeutic loss of function is not beneficial for most diseases and precisely targeted gene addition and gene correction approaches have much lower efficiencies than a simple knockout in vivo. This is particularly true for precise gene editing by homologous recombination[3,40–43]. Here we show that this problem can be overcome by linking a drug resistance scgRNA with a therapeutic transgene in cis within a homologous recombination-dependent GeneRide rAAV vector. After demonstrating the activity of a *Hpd* scgRNA in the powerful *Fah*−/− mouse hepatocyte selection system, we utilized this approach in combination with our recently developed APAP selection method[5]. Fully therapeutic hF9 levels deriving from precisely gene-edited hepatocytes were readily achieved. The initial frequency of hepatocytes corrected by homologous recombination was low (<0.5%) as expected but increased over 100-fold via scgRNA-mediated APAP resistance. While the concentrations of hF9 achieved here are known

to be curative for hemophilia B, hF9 was chosen as a therapeutic transgene for this study primarily because it can serve as a non-invasive secreted biomarker for selection. Most other genetic liver diseases, such as Maple syrup urine disease, Alkaptonuria, Gaucher disease, and Fabry disease, are likely to be amenable to the same selection strategy.

While cell type specific gene editing could theoretically be achieved by the expression of spCas9 from a tissue-specific promoter, spCas9 is much too large to fit into an rAAV vector bearing homology arms and a transgene. Because of its highly recombinogenic single-stranded genome, rAAV is the clear choice as a homology donor for hepatocyte gene targeting. In addition, an RNAP II driven tissue specific scgRNA will permit the use of transient spCas9. The transient presence of spCas9 is preferred as it can prevent off target nuclease activity[44] or immunogenicity[45].

Northern blot analysis and RNA sequencing revealed that the *hF9*-containing fusion transcript from the targeted chromosome was expressed at lower levels (~10-fold) than the native *Alb* message. While this may suggest that the fusion transcripts were either unstable or that the targeted locus is transcribed less efficiently, it is also possible that other technical issues could have prevented the capture of these species for RNA sequencing. To determine whether this effect on mRNA abundance of the target locus is specific to the scgRNA system, a comparison to the previously described selectable GR vectors containing a miR-embedded shRNA was performed[6]. The relative abundance of *hF9* fusion transcripts was compared. Interestingly, the abundance of *hF9* fusion mRNA was very similar in the scgRNA and the miR-embedded shRNA systems. This finding indicates that ribozyme-mediated cleavage of the scgRNA-containing transcript is not likely to be the cause of its lower-than-expected abundance. Future work is needed to determine the molecular causes of this phenomenon and to develop potential remedies. Importantly, however, supratherapeutic levels of *hF9* expression were readily achievable despite the reduced transcriptional strength of the targeted albumin locus.

In the work described here two rAAV vectors are utilized, one for the delivery of the gene targeting GR with the scgRNA and the second for spCas9. To avoid the need for a dual rAAV vector approach and to avoid the potentially deleterious permanent expression of spCas9, future work will use transient expression of spCas9. Transient delivery

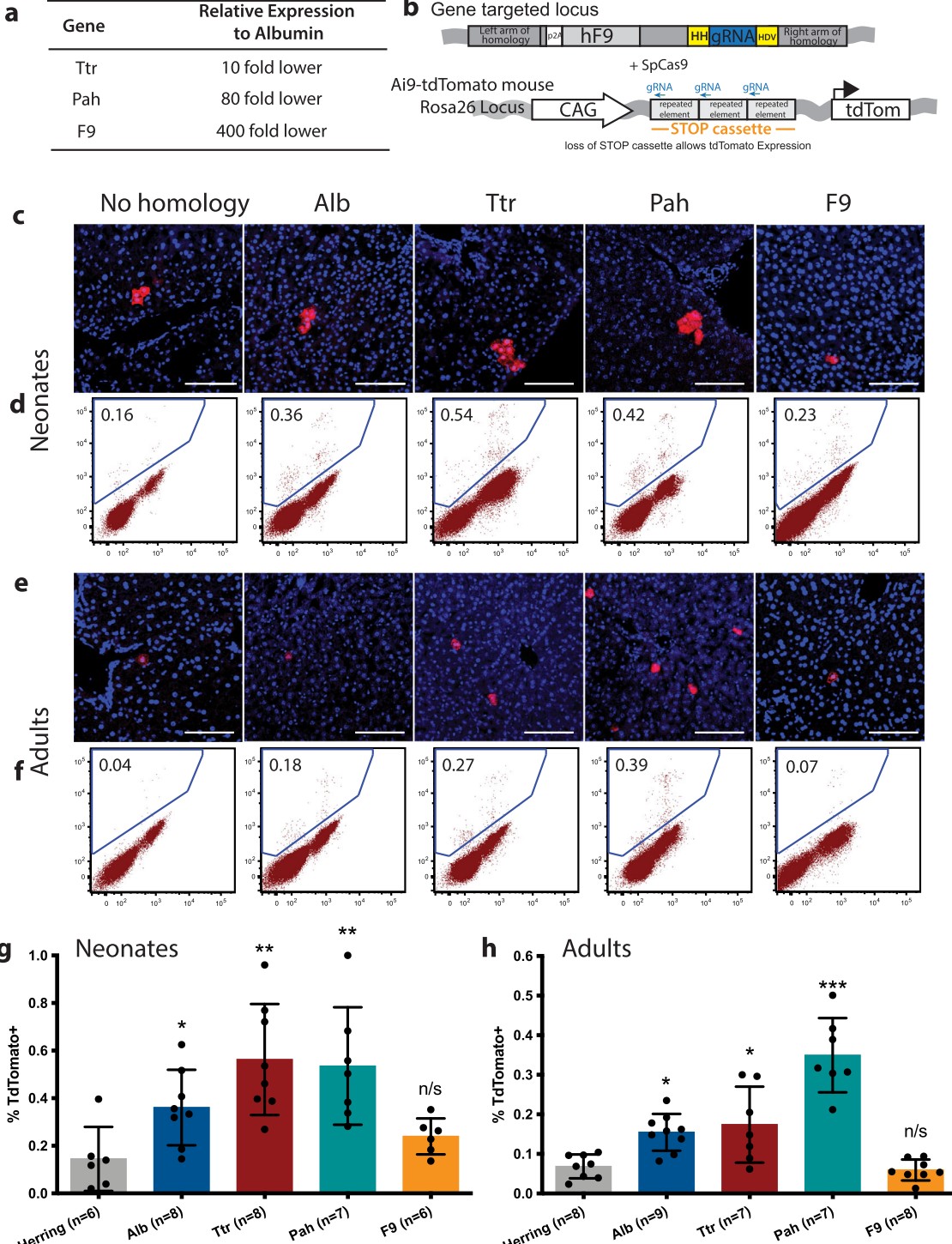

**Fig. 4 | Expression of the GR-scgRNA from weaker expressing genes. a** Table of weaker expressing genes relative to Albumin which were targeted with the GR-stop scgRNA vector. **b** Schematic of a gene targeted locus after homologous recombination with an rAAV-GR vector containing arms of homology to one of the genes listed in (A) and a scgRNA targeting the stop cassette in the Ai9-tdTomato mouse. hF9: human Factor 9; HH: hammerhead ribozyme; gRNA: guide RNA targeting *Hpd*; HDV: hepatitis delta virus ribozyme; tdTom: tdTomato. **c** Representative liver sections and **d** flow cytometry plots showing tdTomato positive hepatocytes in mice that received rAAV-GR targeted to the indicated locus as neonates.

**e** Representative liver sections and **f** flow cytometry plots showing tdTomato positive hepatocytes in mice that received rAAV-GR as adults. Quantification of the tdTomato positive hepatocytes as measured by flow cytometry in mice that receive the various targeting rAAV-GR as **g** neonates and **h** adults. Scale bars represent 100 μm. Biologically independent mice were analyzed. Data are presented as mean values and error bars represent standard deviation. Statistics were calculated using an ordinary one-way ANOVA with Tukey's multiple comparisons test. *$P < 0.05$, **$P < 0.01$ and ***$P < 0.001$. Source data are provided as a Source Data file.

**Table 1 | Integration PCR primers**

|  | Forward primer | Reverse primer |
|---|---|---|
| 1st Alb PCR | 5'-TCCACGAAGAAGCCATGCAG –3' | 5'- CAGATGGTGATCAGGCCAGG-3' |
| 2nd Alb PCR | 5'-AACGTCATGGGTGTGACTTTT –3' | 5'-GGGTTCTCTTCCACGTCGC –3' |
| 1st Ttr PCR | 5'- AACTTGAAAAAGTGGCACCG-3' | 5'-GGCCCTCAAATGAGTAAAGT –3' |
| 2nd Ttr PCR | 5'- TTTCTGATGAGTCCGTGAGG-3' | 5'-CGGATAGGCCCATTCTTGT –3' |
| 1st Pah PCR | 5'-ACCTTCTCAGGTAACTAACG –3' | 5'-CTTAAGTGCAACAAGAGAGAG –3' |
| 2nd Pah PCR | 5'- ATCAACTTGAAAAAGTGGCA-3' | 5'-CGTGCGCATGTGTGT –3' |
| 1st F9 PCR | 5'-ACACTTCCAGCCAGAGGTTAG –3' | 5'-CGGCCATAATCATGTTCACGC –3' |
| 2nd F9 PCR | 5'-ATGTCGTCATACAGGGCCAGA-3' | 5'-GGGTTCTCTTCCACGTCGC-3' |

of spCas9 by lipid nanoparticles[46], nanoclews[47], or ultrasound-mediated gene delivery of microbubbles[48] is desirable.

A potential concern for the safety of the APAP selection system is partial loss of hepatic Cypor activity. A notable safety feature of the system is that Cypor deficiency created by APAP selection is limited to only hepatocytes in zone 3 of the hepatic lobule due to zone 3 specific expression patterns of the Cyp enzymes (Cyp2E1, Cyp1A2) that mediate the metabolism of APAP to NAPQI. This anatomic restriction prevents the possibility that APAP selection would result in a complete loss of hepatic Cypor activity. In our previous publication[5] we demonstrated the long-term safety of partial Cypor deficiency in mice that had been APAP selected. Blood chemistry panels performed at terminal harvest found no significant differences between APAP-selected animals and naïve controls. In mice that had received the GR-*Cypor* scgRNA here and were APAP selected there was mild lipid accumulation in the liver, similar results to those previously observed in Vonada et al.[5]. However, the lipid accumulation seen is much less severe than that described in mice with a complete loss of hepatic Cypor[49], likely due to zone 1 and 2 Cypor activity.

In addition to the *Alb*-targeted GeneRide vector used in gene therapy applications, the feasibility of modulating therapeutic gene expression and the expression of a scgRNA from lower expressing loci was explored through using the Ai9-tdTomato reporter system. This strategy allows the determination of whether the promoter is strong enough to drive the scgRNA as well as the frequency of homologous recombination in a specific locus. TdTomato positive hepatocytes were observed from the targeting of both *Ttr* and *Pah*, which are respectively 1-log and 2-logs lower expressing than *Alb*. Interestingly, there is no correlation of the relative promoter strength to the percentage of hepatocytes expressing tdTomato. One factor affecting the difference in integration frequency at different loci could be the differences in the arms of homology themselves. For lower expressing genes, it is clearly advantageous to use the scgRNA-mediated knockout instead of a shRNA-mediated knockdown. The scgRNA system requires only one effective double strand break to achieve loss-of-function, whereas an shRNA requires a continuous and sufficiently high level of expression to knockdown the targeted gene. The wide dynamic range of the scgRNA system will allow drug selection of precisely gene-modified hepatocyte loci for a wide variety of genetic liver diseases. We were unable to clearly delineate the lower end of the dynamic range of the system, because it was unclear whether functional scgRNA was produced from the *F9* gene, the lowest expressing gene targeted. Even the vector lacking homology arms to the mouse genome produced a low level of reporter gene activation, either due to random integration next to cellular promoters or scgRNA expression from rAAV episomes.

These data show that expression of the scgRNA in a homology-directed integrating vector from an endogenous RNAP II promoter is amenable for the selection and expansion of correctly targeted hepatocytes. The scgRNA system is a useful tool for the expression of a gRNA from an RNP II promoter and maintaining its functional integrity.

## Methods

### Animal husbandry
All animal experiments were approved by the Institutional Animal Care and Use Committee at the Oregon Health & Science University and all experiments were performed according to the guidelines for animal care at Oregon Health & Science University. Wildtype C57BL/6, B6J-Rosa-Cag-spCas9 (Stock #026179), and B6-Rosa-Cas9-Ai9tdTomato (Stock #007909) mice were obtained from Jackson Laboratories. *Fah*⁻/⁻ mice were of the FahΔexon5 strain on the 129s4 background[50] and were maintained on drinking water containing 8 mg/L NTBC (Ambeed). Mice were housed in groups of 2 to 5 animals in the Department of Comparative Medicine at the Oregon Health and Science University. The animal room was maintained at a temperature of 70 °F and 30–70% humidity. All animals were fed tap water and standard mouse chow (LabDiet Picolab Rodent Diet 5LOD). Animals were housed under a standard 12 h on and off light cycle. Mice were euthanized with $CO_2$. A cardiac puncture was performed immediately after death to deplete the liver of blood.

### Vector construction
The scgRNA targeting *Hpd* and *Cypor* were synthesized and ligated into the Albumin-targeting GeneRide rAAV vector[6] using the AflII and AgeI restriction sites. Ribozyme sequences used for the scgRNA were previously described[16]. The *Hpd* gRNA (5'-GACGTGGCTGACTACCTCCC-3') was designed using http://crispor.tefor.net/[13]. The *Cypor* gRNA (5'-TCGTGGGGGTCCTGACCTAC-3') used has previously been validated[5]. The rAAV8-LSP-spCas9 vector consists of the tiny liver specific promoter (LSP)[51] (a gift from Cary Harding) driving the expression of spCas9. spCas9 was restriction digested from the pX330-U6-Chimeric_BB-Cbh-hSpCas9 plasmid (a gift from Feng Zhang, Addgene plasmid #42230; http://n2t.net/addgene:32230; RRID: Addgene_42230) using AgeI and EcoRI and ligated into an rAAV vector plasmid. The LSP promoter was PCR amplified and then ligated into the NotI site on this plasmid. rAAV serotype 8 was produced from this final plasmid. Arms of homology for the murine *Ttr* (left arm of homology chr18:2-672690-20673738; right arm of homology chr18: 20673739-20674830), *Pah* (left arm of homology: chr10:87582529-87583578; right arm of homology: chr10:87583579-87584678) and *F9* (left arm of homology: chrX:60028079-60029441 and right arm of homology chrX: 60029442-60030474) were designed such that the scgRNA would be integrated downstream of the terminal coding exon. Arms of homology were PCR amplified from C57BL/6 mouse genomic DNA using Phusion High-Fidelity DNA Polymerase (New England Biolabs), and the inserts were cloned into an rAAV backbone[52] using the InFusion Cloning Kit (Takara Bio). The DNA with no homology to mouse genomic DNA was cloned from the Atlantic Herring evolv1b gene by PCR amplification of herring sperm DNA (left arm chr NC_045161.1: 23619366-23620603 and right arm chr NC_045161.1: 23620604-23621784). The gRNA targeting the stop sequence upstream of tdTomato in the Ai9-tdTomato mouse (5'-AAACCTCTACAAATGTGGTA-3') was chosen from a previously published list of gRNAs targeting the stop sequence[31].

The Ttr targeting plasmid containing luciferin and a Cypor scgRNA was constructed by taking the GR-stop scgRNA plasmid with arms of homology to Ttr and restriction digesting it with XhoI and AgeI to remove the hF9 transgene. Phusion high fidelity polymerase was used to amplify luciferase from the pGL3-Basic-Luciferase plasmid (gift from Philip Streeter, OHSU). The luciferase construct was inserted it into the plasmid using the InFusion Cloning Kit (Takara Bio). The plasmid was then restriction digested with AgeI and HindIII. Cypor-scgRNA was synthesized and ligated into this cloning site. Vector design was performed using Snap Gene version 6.1. All plasmid vectors that are not publicly available can be obtained from the authors upon request.

## rAAV production and delivery

To generate rAAV, the rAAV plasmid vector was packaged in the AAV-DJ or AAV-8 serotype by cotransfection of HEK293 cells with the vector plasmid, pDJ or pAAV8 (capsid plasmid), and pHelper. $7 \times 10^6$ or $20 \times 10^6$ HEK293T cells were plated into 10 cm or 15 cm dishes and cultured in DMEM complete media containing 10% fetal bovine serum (Hyclone), 100 U/mL penicillin (Gibco), 100 µg/mL streptomycin (Gibco), 2 mM GlutaMax (Gibco) and 1 mM sodium pyruvate (Gibco). After 16–20 h, the media was replaced with reduced serum DMEM containing 2% fetal bovine serum. Cells were transfected by polyethylenimine (PEI) based triple transfection. Equimolar quantities of each plasmid were combined to a total mass of 24 µg in OptiMEM (Gibco) media and 3 times the plasmid weight of PEI (72 µg from a 1 mg/mL stock solution, pH 7.1 (Polysciences)) was added, with a final volume of 1.5 mL per 10 cm dish or 4.5 mL per 15 cm dish[53]. The transfection mixture was incubated at room temperature for 20 to 30 min and then added to the cells. 5 days post transfection the media was collected and centrifuged at $1900 \times g$ for 30 min, 2 times. The supernatant was treated with benzonase (Sigma) at 37 °C for 1 h then centrifuged at $1900 \times g$ for 30 min. The supernatant was collected and filtered through a 0.45 µm PES vacuum filter and precipitated with 0.8% PEG 8000 (w/v) (Spectrum Chemical) and 0.5 M sodium chloride (Fisher Scientific) overnight at 4 °C. The following day the supernatant was centrifuged at $6000 \times g$ for 1 h. The supernatant was then removed and discarded and the viral pellet was resuspended in sterile 1× PBS. rAAV titers were determined by P32 radioactive dot blots. All rAAV vectors mixed with saline were delivered to P0–P1 neonates of both sexes via facial vein injection in a total volume of 14 µL per pup. $1 \times 10^{11}$ vg/pup or approximately $1 \times 10^{14}$ vg/kg of each of the GR-*Hpd* scgRNA rAAV and GR-*Cypor* scgRNA rAAV vectors were administered to neonatal mice. Both these GR vectors were of the AAVDJ serotype. At weaning a subset of these mice received $1 \times 10^{11}$ vg per mouse or approximately $8 \times 10^{12}$ vg/kg of rAAV8-LSP-spCas9. For the weaker gene expression experiments, $8 \times 10^{10}$ vg of each GR-stop scgRNA rAAV was diluted in saline to a total volume of 14 µL and injected via facial vein injection into P0–P1 mice. Delivery of GR-stop scgRNA rAAV to 3- to 10-week old adult mice of both sexes was performed by retro-orbital injection. For these mice, $6 \times 10^{11}$ vg per mouse of each of the GR-stop scgRNA rAAV targeting vectors was delivered into 8- to 10-week old mice, in a total volume of 50–100 µL.

## Reverse transcription and quantitative real time PCR (qRTPCR)

Cellular RNA was extracted from liver tissue using RNAzol RT (Molecular Research Center Inc.) following the manufacturer's protocol. Random hexamers were used with SuperScript IV First-Strand Synthesis System (Invitrogen) to synthesize cDNA. qRTPCR was performed utilizing FastStart Essential DNA Green Master Mix (Roche) on a Lightcycler 96 SW version 1.1 (Roche). *Hpd* was amplified with forward primer 5′-TGGAATGTAAGCCCCCATCC-3′ and reverse primer 5′-AGCC AGTTGGAGATGACTGATG-3′. *Gapdh*, used to normalize gene expression levels, was amplified with forward primer 5′-AAGGTCGGTGT GAACGGATTTGG-3′ and reverse primer 5′-CGTTGAATTTGCCGTGA GTGGAG-3′.

## Acetaminophen selection

For Acetaminophen (APAP) selection mice were fasted for approximately 16 h prior to acetaminophen administration. The mice received biweekly intraperitoneal injections of 13 mg/mL acetaminophen (4-Acetamidophenol, 98%, ACROS Organics) in saline[5]. APAP was dissolved in prewarmed saline and sterilized with a 0.2 µM filter prior to use. Doses of acetaminophen were started at 220 mg/kg in males and 250 mg/kg in females and increased by 5–10 mg/kg in subsequent doses until an elevated ALT (>800 IU/liter) response was observed. For analysis of an ALT response, blood was taken 6–7 h post APAP injection. Animals were monitored for potential adverse effects 6, 24, 48, and 72 h post APAP injection.

## Blood chemistry

Blood samples for both the hF9 and ALT assays were collected from the saphenous vein at each of the indicated timepoints. hF9 was measured using the Asserachrom IX:Ag ELISA kit (Stago) following the manufacturer's protocols with modifications based on a previous publication[6] as follows. 5 µL of blood sample were collected from the saphenous vein puncture and diluted into 245 µL of R4 buffer from the ELISA kit supplemented with 1.1 mM sodium citrate (Fisher Scientific) as an anticoagulant. Samples were centrifuged at $5000 \times g$ for 5 min in a table top centrifuge and supernatants were stored at −80 °C until ready for analysis (up to 6 weeks). 200 µL of each sample was used for the hF9 ELISA. The limit of detection was about 20 ng/ mL hF9 in whole blood. To measure blood Alanine Transaminase (ALT), 10 µL of blood was drawn and placed into 15 µL of saline containing heparin. Samples were centrifuged for 3 min and supernatant was collected and stored at −20 °C until ready for analysis (up to 7 days). Blood ALT levels were measured in quadruplicate using the colorimetric ALT (SGPT) Color Endpoint kit (TECO Diagnostics), following manufacturer's protocol. Analysis of the assays were performed using Gen5 3.03 software.

## TIDE analysis of insertion and deletion mutations

Insertion and deletion mutations were quantified with the assumption that hepatocytes comprise 60% of the liver[54]. Harvested livers were homogenized in saline at a 1:1 ratio and stored at −80 °C until extraction was performed. Genomic DNA was extracted from homogenized liver tissue using the MasterPure Complete DNA and RNA Purification kit (Lucigen) following the manufacturer's instructions. The region surrounding the *Cypor* gRNA cut site (Forward primer: 5′-GTTT GCGGGTGTTAGCTCTTC-3′, Reverse primer: 5′-AGTCTACTTCAGTC GCAGCC-3′) and *Hpd* gRNA cut site (Forward primer: 5′-GAACTGGGAT TGGCTAGTGC-3′, Reverse primer: 5′-GCCCTTCCCTACATCCTAGT-3′) were amplified with MyTaq Red Mix (Bioline). PCR products were then purified using the PCR clean-up and gel extraction kit (Macherey-Nagel) and Sanger sequenced. The forward primers served as sequencing primers. Indels for each mouse were analyzed using the TIDE software, https://tide.nki.nl[26].

## Histology

Upon harvest, tissue was fixed in 4% paraformaldehyde (Sigma-Aldrich) for 4 h at room temperature or overnight at 4 °C. Tissues were prepared for cryopreservation by passing through a 10, 20, and 30% sucrose in PBS (w/v) gradient. Tissues were then embedded in OCT (Tissue-Tek) and cut into 7–8 µm sections using a cryostat onto Colorfrost Plus slides (Fisher Scientific). All liver tissue sections were treated with TrueBlack Lipofuscin AutoFluorescence Quencher (Biotium) at a 1:200 dilution in 1×PBS for 1 minute then washed 3 × 5 min in 1×PBS prior to any further manipulation. Sections were permeabilized in 0.1% Triton X-100 in 1×PBS at room temperature for 12 min, washed 3 × 5 min in 1×PBS, and then blocked in 10% normal goat or donkey serum with 0.3 M glycine for 30 min at room temperature. Slides were incubated with primary antibody diluted in 10% normal donkey serum

overnight at 4 °C. Primary antibodies used include rabbit anti-Cypor (Abcam, catalog #180597, dilution 1:200) and goat-anti-human F9 (Affinity Biologicals, catalog # GAFIX-AP, dilution 1:100). Slides were washed 3 × 5 min in 1×PBS then incubated in secondary antibody diluted in 10% normal donkey serum for 30–90 min at room temperature. Secondary antibodies used include Alexa-Fluor 647 donkey-anti-rabbit IgG (Jackson ImmunoResearch catalog #711-606-152, dilution 1:400) and Cy3 donkey-anti-goat IgG (Jackson ImmunoResearch, catalog #705-165-147, dilution 1:400). All tissue sections were stained with Hoescht 33342 (Invitrogen) at 1:10,000 dilution in 1×PBS for 3 min then washed 3 × 5 min in 1×PBS. Coverslips were mounted with Fluoromount-G (SouthernBiotech). Imaging was performed on a Zeiss LSM700 confocal microscope using Zeiss Zen Software version 2.3 Blue Edition (Carl Zeiss AG). Image analysis was conducted manually using ImageJ version 1.52k.

## Hepatocyte isolation and flow cytometry

Prior to perfusion, liver perfusion solution 1 was prepared by adding 10 mM HEPES (Fisher Scientific) and 0.5 mM EGTA (Fisher Scientific) to calcium and magnesium-free Earle's Balanced Salt Solution (Gibco) and liver perfusion solution 2 was prepared by adding 5 mM EDTA (Sigma) to calcium and magnesium-containing Earle's Balanced Salt Solution (Corning). Both solutions were stored at 4 °C until ready the day of perfusion. 0.1 mg/mL collagenase type II solution was prepared by dissolving the collagenase type II (Worthington Biochemical) in calcium and magnesium-containing Earle's Balanced Salt Solution and stored at −80 °C until ready to use.

For hepatocyte isolation, all perfusion solutions were prewarmed and maintained at 37 °C for the duration of the perfusion. Mice were anesthetized by administering a cocktail consisting of 75 mg/kg Ketamine, 2.5 mg/kg Acepromazine, and 15 mg/kg Xylazine by intraperitoneal injection. Once anesthetized, the abdominal cavity was opened and the portal vein was cannulated with a 24G × ¾" Terumo Surflo I.V. catheter (Terumo). The liver was perfused with liver perfusion solution 1 at 4 mL/min using a peristaltic pump (Ismatec) until the liver was blanched. An incision was then made to the inferior vena cava and 1 mg of collagenase type II solution (10 mL of 0.1 mg/mL stock solution) was added to liver perfusion solution 2. The perfusion solution was the switched to liver perfusion solution 2. Once fully digested (as determined by visible indentation and cell dissociation upon lightly contacting the liver with a cotton applicator), the liver was removed from the abdominal cavity of the mouse and placed into a 60 mm sterile petri dish with 3–5 mL liver perfusion 2 solution containing collagenase type II. Gentle, manual disruption of the liver capsule was performed with forceps. Dulbecco's minimal essential medium (DMEM) (Gibco) containing 10% fetal bovine serum (FBS) (Hyclone) was added to the petri dish and the hepatocytes were filtered through a 100 μm cell strainer (Fisher) and followed by a 70 μm cell strainer (Fisher) into a 50 mL tube (Falcon) to remove any large debris. Collected hepatocytes were then washed and resuspended in DMEM containing 10% FBS and centrifuged twice at 50 × $g$. Isolated hepatocyte samples in DMEM containing 10% FBS were analyzed on a Symphony flow cytometer (Becton Dickinson) running FACS Diva version 8.0.2 or a Fortessa (Becton Dickinson) running FACS Diva version 9.0 and FlowJo version 10.6.1 (Treestar) software program.

## Northern blotting

RNA was extracted by homogenizing liver tissue samples in RNAzol (Molecular Research Center, Inc.) and following the manufacturer's protocol for extraction. Extracted RNA was resuspended in Formamide (Fisher Scientific). RNA concentration was determined by Qubit (Invitrogen). 6 μg of total RNA per sample was prepared for electrophoresis by adding 25 μL of sample buffer and ddH₂O to a total volume of 30 μL. To prepare 25 μL sample buffer mix together 2.5 μL

10× MOPS buffer (0.2 M MOPS, 80 mM sodium acetate·3H₂O and 10 mM EDTA, pH 7.0 adjusted with NaOH), 12.5 μL formamide (Fisher Scientific) and 5 μL 37% formaldehyde (Fisher Scientific). Samples in sample buffer were heated to 65 °C for 5 min then placed on ice. 5 μL of loading dye containing 1% of 1 mg/mL ethidium bromide (Invitrogen) was added to each sample[55]. Samples were electrophoresed in a 1.0% or 1.5% agarose (Thermo Fisher Scientific) gel containing 2% formaldehyde (Fisher) in a running buffer of 1×MOPS. Gels were then soaked in 20×SSC (3 M sodium chloride (Fisher Scientific), 300 nM sodium citrate (Fisher Scientific), pH 7.0) for 30 min and RNA was transferred overnight to a Hybond-N+ membrane (GE Healthcare). Following the transfer, the membrane was UV cross linked and incubated in prehybridization buffer for 1 h at 42 °C. Prehybridization buffer consists of 50% formamide (Fisher Scientific), 5× SSPE (Thermo Fisher Scientific), 2× Denhardts solution (100× Denhardts solution contains 2% Ficoll 400 (Sigma-Aldrich), 2% polyvinyl pyrrolidone, 2% bovine serum albumin (Sigma) in ddH₂O), 0.1% sodium dodecyl sulfate (SDS; Fisher Scientific) in ddH₂O. While the membrane was incubating, 7.2 μg/mL herring sperm DNA (10 mg/mL; Thermo Fisher Scientific) was denatured by heating to 75 °C for 10 min then added to the prehybridization buffer and incubation was continued at 42 °C for an additional 1 h. Probes for mouse *Alb* and *Ttr* were generated by PCR with the following primers: *Alb* forward primer 5′-ACACATGTACT TGGCAAGTT-3′ and reverse primer 5′-CAGACACACACGGTTCAGG AT-3′; *Aat* forward primer 5′-ACAAACATCGGAGGCTGACA-3′ and reverse primer 5′-TGGTCAGCACAGCCTTATGC-3′. The probe for *hF9* was generated by restriction digest of the rAAV plasmid with SpeI and AflII to create a 1099 base pair fragment. Probe was boiled for 1 minute and added to the prehybyridization buffer with herring sperm DNA. Membrane was incubated at 42 °C with rotation. The following day, the membrane was rinsed in wash buffer consisting of 1×SSC with 0.1% SDS for 5 min. The membrane was then washed in wash buffer for 2 h at room temperature and exposed for visualization.

## RNA Sequencing and data analysis

RNA library preparations and sequencing reactions were conducted at GENEWIZ, LLC. Sequencing results are from mice that received the GR-*Cypor* scgRNA as neonates with 3 APAP selection mice and 3 unselected mice. RNA samples were quantified using Qubit 2.0 Fluorometer (Life Technologies) and RNA integrity was checked using Agilent TapeStation 4200 (Agilent Technologies).

RNA sequencing libraries were prepared using the NEBNext Ultra RNA Library Prep Kit for Illumina following the manufacturer's instructions (New England Biolabs). Briefly, mRNAs were first enriched with Oligo(dT) beads. Enriched mRNAs were fragmented for 15 min at 94 °C. First strand and second strand cDNAs were subsequently synthesized. cDNA fragments were end repaired and adenylated at 3′ends, and universal adapters were ligated to cDNA fragments, followed by index addition and library enrichment by limited-cycle PCR. The sequencing libraries were validated on the Agilent TapeStation (Agilent Technologies), and quantified by using Qubit 2.0 Fluorometer (Invitrogen) as well as by quantitative PCR (KAPA Biosystems).

The sequencing libraries were clustered on flowcells. After clustering, the flowcell was loaded on the Illumina HiSeq instrument (4000 or equivalent) according to manufacturer's instructions. The samples were sequenced using a 2 × 150 bp Paired End (PE) configuration. Image analysis and base calling were conducted by the HiSeq Control Software (HCS) version HD 3.5.0.7. Raw sequence data (.bcl files) generated from Illumina HiSeq was converted into fastq files and demultiplexed using Illumina's bcl2fastq 2.17 software. One mismatch was allowed for index sequence identification.

For the custom alignments, Bowtie version 1[56] was used to create custom indexes and align data from knock in mouse model. Indexes were made from selected knock in sequence and transcripts of targeted genes. Default parameters were used to build indexes. A 50 base

wide window (shifted in steps of 15 bases) was used to go through the sequenced data looking for alignments and selected the first valid alignment found. For valid alignments, 3 mismatches were allowed, but required unique best matches (–best–strata -v 3 –m 1). Once all alignments were found, full sequences were stacked based on positions of windowed alignment. This allowed us to see the "overhang" (in an unbiased way) where our knock in sequence was inserted into the host genome (mouse genome assembly GRCm38). Paired end sequencing was performed, but aligned to pair ends independently. The alignment strategy did not generate many mated pairs and mated pairs were not required, but when counting alignments, mated pairs were only counted once.

For the TPM analysis in supplemental Fig. 3c, d, paired-end fastq sequences were trimmed using trim-galore (ver 0.6.3) and default parameters. Pseudoalignment was performed with kallisto (ver 0.44.0) using genome assembly GRCm38 and gencode (ver 24) annotation; default parameters were used other than the number of threads. The Bioconda package bioconductor-tximport (ver 1.12.1) was used to create gene level counts and abundances (TPMs). Quality checks were assessed with FastQC (ver 0.11.8) and MultiQC (ver 1.7). Quality checks, read trimming, pseudoalignment, and quantitation were performed via a reproducible snakemake pipeline, and all dependencies for these steps were deployed within the anaconda package management system[57–60].

### Integration PCR
For mice that received the GR-*Hpd* scgRNA and the GR-*Cypor* scgRNA, PCRs using MyTaq Red Mix (Bioline) were performed to show correct integration of the targeting rAAV vectors into the Albumin locus. For these PCRs the primers called 2nd Alb PCR from the table below were used. For mice that received the GR-Stop scgRNA with different homology arms, nested PCRs were performed to detect integration events using MyTaq Red Mix (Bioline). Both sets of PCR primers were designed to span the integration junction with one primer outside of the arm of homology and one primer in the hF9 or P2A transgenes. Nested PCRs were performed by diluting the initial PCR product 1:100 and using 1 μL in the subsequent nested PCR. Primers used are listed in Table 1. Samples were electrophoresed on a 1.2–1.4% agarose gel (Thermo Fisher Scientific) with a Gene Ruler 1 kb plus DNA ladder (Thermo Fisher Scientific) for visualization.

### Western Blot
Liver samples were homogenized in HEN buffer (10 mM HEPES pH 7.4, 10 mM EDTA pH 8.5, 25 mM NaCl, 1 mM PMSF and 1× complete protease inhibitor cocktail). 1% Triton X-100 was added and supernatant was removed and stored at −80 °C until ready to use. Samples were run on a 4–12% Bis-Tris SDS-polyacrylamide gel (Invitrogen) and transferred to a PVDF membrane overnight. The membrane was blocked in 5% nonfat dry milk (NFDM) in DPBS for 2 h at room temperature and probed with either rabbit-anti-mouse Albumin antibody diluted 1:1000 (Cell Signaling, Catalog #4929) or Gapdh antibody diluted 1:1000 (Cell Signaling, Catalog #2118) in 5% NFDM in DPBS with 0.1% Tween-20 (PBST) overnight at 4 °C. The membrane was then washed in PBST 3 times and place in horseradish peroxidase conjugated secondary antibody diluted 1:10,000 (Prometheus protein biology products, Catalog #20-304D) in 5% NFDM in DPBS at room temperature for 2.5 h. The membrane was then detected by chemiluminescence using a ChemiDoc Touch Imaging System (BioRad) for a visualization.

### In vivo imaging
30 mg/mL of D-luciferin (Fisher scientific) in saline stock was made. Mice were shaved in the abdominal area prior to imaging. 15 min prior to imaging, mice were given luciferin in saline at a dose of 150 mg/kg. In vivo imaging was performed on the IVIS Lumina II.

### Statistical analysis
All data is presented as mean ± standard deviation. All statistical analysis was performed using GraphPad Prism version 9.4.0 for Mac, GraphPad Software, La Jolla California, www.graphpad.com. Experimental differences were evaluated by a two-tailed unpaired Student's t test in Fig. 1, Fig. 2, and part of Fig. 3. A one-way ANOVA with Tukey's multiple comparison test was used to determine the experimental differences of the flow cytometry results in Fig. 4. *P*-values <0.05 were considered statistically significant.

### Reporting summary
Further information on research design is available in the Nature Portfolio Reporting Summary linked to this article.

## Data availability
The authors declare that all data supporting the finding of this study are included in this published article, supplemental materials, the Source Data file. Source data are provided with this paper. The reference mouse genome assembly used is GRCm38 and annotation was made by using gencode version 24. The RNA sequencing data generated in this study have been deposited in the Gene Expression Omnibus database under the accession code GSE216550. Source data are provided with this paper.

## Code availability
The custom code used in this study has been deposited in the Zenodo repository with DOI: 10.5281/zenodo.7268404.

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

## Acknowledgements

We thank A. Major at Baylor University for her help with tissue histology and Hannah Holly at the Oregon Health and Science University Knight Cancer Institute for providing the pipeline for the analysis of RNA sequencing for gene expression. A.T. is a Ruth L. Kirchstein National

Research Service Awards Postdoctoral Fellow (NIDDK, 1 F2 DK117516-01). M.G. is supported by the NIH (5R01DK123093). This research was sponsored in part by a research agreement with LogicBio Therapeutics.

## Author contributions

A.T. planned, designed and performed the experiments, analyzed the data, generated figures, and wrote the manuscript. M.G. designed the experiments and wrote the manuscript. A.V. and J.P. performed experiments. J.P. produced rAAV vectors and quantified rAAV titers. L.A.W. assisted with animal husbandry. C.P. wrote the code and performed the bioinformatics analysis.

## Competing interests

The authors declare that Oregon Health and Science University, M.G., A.T., and A.V. have a financial interest in LogicBio Therapeutics, a company that may have a commercial interest in the results of this research and technology. M.G., A.T., and A.V. are inventors of technology that is utilized in this research, and that has been licensed to LogicBio Therapeutics. M.G. has significant financial interests in Yecuris Corp. and Ambys Medicines. A patent application on the APAP selection technology described herein has been filed by OHSU (Title: Methods of Gene Therapy. Filing number: PCT/US19/29890. M.G. and A.T. are coinventors). These potential conflicts of interest have been reviewed and managed by the Oregon Health and Science University. The remaining authors declare no competing interests.
