## [Peer Review File · Nature Communications]

Self-cleaving guide RNAs enable pharmacological selection of precise gene editing events in vivoREVIEWER COMMENTS

Reviewer #1 (Remarks to the Author):

In this paper, the team of Grompe use self-cleaving guide RNA (scgRNA) using ribozyme motifs to create drug-selectable gene targeting events in hepatocytes. To achieve this, the authors deliver an AAV vector containing a promoter-less therapeutic gene flanked with homology regions that enable targeted integration in the albumin locus, co-opting this endogenous promoter for its expression. In addition, the same AAV construct also contains the scgRNA-encoding gene. Consequently, the scgRNA is also integrated into the albumin locus and is expressed from the endogenous albumin promoter. Hence, this approach allows for tissue-specific expression of gRNA obviating the need to rely on aspecific, ubiquitously expressed RNA pol III promoters (such as U6). The scgRNA was designed to target genes that when targeted confer a selective survival advantage to the gene-modified hepatocytes in the presence of a selective drug. This was validated in the FAH^{-/-} tyrosinemia mouse model after selective targeting of the Hpd gene by scgRNA and Cas9 and upon withdrawal of NTBC. An alternative model was also used based on acetaminophen-selection of hepatocytes in which the cytochrome p450 reductase (Cypor) had been inactivated by scgRNA/Cas9. Using 2 different models significantly strengthens the study.

Given the low efficiency of gene targeting by homologous recombination in hepatocytes there is an unmet need in the field to select for gene-edited cells *in vivo*. The current data establish proof of concept that scgRNA can be used for selective enrichment of gene-edited hepatocytes *in vivo* resulting in therapeutic expression levels of a gene of interest (e.g. factor IX). To my knowledge, this is the first report that exploits the scgRNA-mediated gene targeting and the use of Cypor as a targeting strategy for pharmacological acetaminophen-mediated *in vivo* selection of gene-edited hepatocytes with therapeutic consequences. *In vivo* selection of gene-edited hepatocytes in the FAH model is well established but the development of *in vivo* selection strategies that go beyond FAH significantly broaden the scope of applications in gene editing. Though the authors previously used the same approach to select for lentivirally-transduced hepatocytes *in vivo*, the current study goes significantly beyond this by demonstrating efficient *in vivo* selection of gene-targeted hepatocytes that are in limited abundance compared to lentivirally transduced cells. The methodology is sound and the findings are novel.

Comments

1. What are the long-term consequences and safety aspect of inactivating Cypor. This should preferably be addressed experimentally after selection of the Cypor-negative cells. Transgenic mouse studies based on selected inactivation of Cypor indicate significant increases in liver weight, accompanied by hepatic lipidosis, abnormal cholesterol homeostasis and other pathologic changes, especially long-term (after 2 months) (Gu et al., J Bioch Chem. 278:25895, 2003).

2. I appreciate that hemophilia is used as a model to establish proof of concept. However, since long-term efficacy and safety has been established following conventional AAV-based gene therapy (not gene editing) in severe hemophilia B patients in clinical trials, the case for using this approach for this particular disease is not very convincing. Nevertheless, the targeting and selection approach may have merit in pediatric subjects to overcome the limitation of diluted AAV genomes upon hepatocyte cell division. However, the substantial cell death that occurs following administration of toxic doses acetaminophen is a concern in disease like hemophilia for which there are other treatment options and given the advances in conventional gene therapy. This strategy would ultimately be better suited for severe lethal diseases of high unmet medical need for which there are no effective therapies available. This should at least be discussed with a clear focus on which diseases may ultimately benefit from this strategy and for which the hepatotoxicity following *in vivo* selection would be considered acceptable.

3. The authors employ a promoter-less construct to achieve targeting expression of the therapeutic gene based on prior work previously published by M. Kay's group (Barzel et al. Nature 517: 360-4, 2015). The merit of that system (designed as "GR" or "gene-ride" in the manuscript) is that it enables

gene targeting in the absence of CRISPR (or any other nuclease-based strategy to achieve targeted double strand DNA breaks) to circumvent possible limitations of using CRISPR for gene targeting (i.e. off-target effects, immune clearance of gene-targeted cells due to the immunogenicity of Cas9 etc.). Hence, the CRISPR Cas9-independence is the main attribute and “sales-pitch” of this GR platform technology. However, in the current study, CRISPR Cas9 was introduced into the hepatocytes to achieve targeted scgRNA-mediated disruption of the Hpd or Cypor genes in order to confer a selective advantage to the gene-edited cells. Consequently, the main advantage of using the GR platform in the first place is eclipsed by the reintroduction of the CRISPR system into the target cells. In this case it is not clear what the main advantage is of relying on the GR platform, given the low overall efficiency of CRISPR-independent targeting in the albumin locus. Since it is expected that the targeting efficiency of the promoter-less construct can be increased using CRISPR/Cas9 and since Cas9 is already present in the hepatocytes, it would make more sense to assess the potential for multiplexing using scgRNA that are specifically designed to target: (i) the Hpd or Cypor genes to enable selection of gene-edited hepatocytes and (ii) the albumin locus to enhance the efficiency of targeted integration of the FIX gene downstream of the endogenous albumin promoter.

4. The authors claim that one of the main advantages of the platform is that it enables expression of the gRNA in a tissue-specific manner by co-opting the endogenous RNA polymerase II albumin promoter upon targeted integration as opposed to using ubiquitously expressed RNA pol III promoters to drive the gRNA expression (like U6 or H1). The concept of using pol II promoter to drive scgRNA has been published previously and prior publications were cited by the authors. In addition, the scgRNA concept has been used for targeted upregulation of therapeutically relevant proteins (e.g. Xu et al. *Nucleic Acids Res* 2017 Mar 17;45(5):e28). The advantage of expressing the scgRNA in a cell type-specific manner in vivo to achieve cell type specific targeting is not a gamechanger since this has been made possible by expressing the Cas9 protein itself from a tissue-specific promoter, particularly in the liver. What is the promoter used in the current studies to drive Cas9 in the different models? This was not specified.

5. Fig S3: Though TTR is a weaker promoter than albumin it is still considered a very strong hepatocyte-specific promoter that has therefore been used in gene therapy clinical trials. The case for demonstrating the potential of this technology in the context of weaker promoter is not very strong, particularly since the effectiveness of the strategy has not been demonstrated when the FIX endogenous promoter was targeted.

6. It would appear that the FIX levels attained without selection are much lower than what has been published previously (Barzel et al. *Nature* 517: 360-4, 2015). What is the reason for this difference?

7. Do Cypor mRNA levels correlate with the targeting efficacy?

8. Is the gene inactivation of the Cypor and Hpd genes mono-allelic or bi-allelic? What about the targeting at the albumin locus ?

9. It is difficult to reconcile the high levels of FIX protein with the very low FIX mRNA levels. This does not quite add up. It would be interesting to ascertain in a controlled experiment whether the low FIX mRNA levels are due to the incorporation of the scgRNA cassette

10. The authors ascribe fact that the ribozymes cannot be detected due to their instability but there is no evidence to support this. It cannot be ruled out that this is due to a technical issue

11. In the targeting construct, why is there a central region of homology to the target sequence? Does it does suffice to have homology regions at the 5' and 3' end only?

12. The tdTomato mouse constitutively express Cas9. It can be expected that the targeting efficiency would be much lower if clinically relevant AAV-Cas9 vector doses would be administered instead of using a Cas9 transgenic model.

Reviewer #2 (Remarks to the Author):

Tiyaboonchai and colleagues presents the continuation of previous work aimed at pharmacologically selecting hepatocytes genetically modified with the GeneRide platform. In previous work, the authors used RNA mediated selecting to enable drug selection of modified hepatocytes. Here, they use the co-expression of guide RNA to enable CRISPR mediated drug resistance. The work is interesting as it could help improve levels of transgene expression obtained with the GeneRide system.

Major points:

- Give the previous work published by the authors, it is important to clearly outline the points of novelty of the current work. The basic mechanism exploited to allow for hepatocyte selection is in fact the same. One point to carefully address is that the method presented here has potential drawbacks, including the fact that a double strand cut in the genome can lead to genome instability, particularly in a setting of induced toxicity and cell proliferation, and that in large animal models and human the starting efficiency of AAV transduction in liver is remarkably smaller than in mice, thus making the approach more challenging (e.g., requiring prolonged administration of hepatotoxic drugs to achieve therapeutic levels of expression).
- Here two AAV vectors are administered, one containing the promoterless targeting sequence and the scgRNA and the other the Cas9. What is interesting is that the second vector still appears to work, even in the potential presence of antibodies induced by the first vector. Complete characterization of antibody titers and Cas9 expression (compared to naïve mice) is warranted. Also, the authors should point out that the approach is hardly translatable and will require the use of LNPs or other systems.
- Figs 1 and 2. Given the high vector dose used, it would be important to characterize the integration pattern of the GeneRide vector and albumin protein levels. They should also quantify immunohistochemistry images to understand what is driving FIX expression and make sure all readouts correlated.
- Figure 4 is confusing, seems like results do not correlate with the expected expression strength of the genes selected, as for example TTR is better than ALB in this model. Were the mice treated with acetaminophen? If not, it would be important to administer acetaminophen to compare results with Figs 1 and 2 and particularly to compare what happens in the GeneRide vs. no homology groups. The levels of targeting seem also low, are they like those in Fig 1-2? The actual dose of vector was quite similar (8×10^{10} vs. 1×10^{11} vg). As discussed above, antibodies to AAV should be measured in all animals as well as Cas9 expression. For the FIX gene, low expression of the scgRNA and low expression of Cas9 could explain the background levels of TdTomato+ cells.

Reviewer #3 (Remarks to the Author):

The authors report a method that allows to provide a growth advantage to hepatocytes upon correct insertion of a therapeutic gene, FIX, by gene ride. Upon selection the efficacy of the approach is significantly increased providing therapeutic correction. The approach taken follows up on previous studies of this group.

Comments:

The data presented in fig 1 and 2 are straightforward showing efficacy of the strategy.

I do have major problems with figure 3 that need to be clarified and improved.

In 3 A the signal for mAlb is shown. The levels are very different and strangely, no signal is detected in the non-selected animals. Please explain. The quantification given in the table does present very different results and only a limited spread in expression. How was this calculated.

In 3B although only in the selected animals the mRNA with the correct size is found, as expected, again the levels are completely different from no signal to a clear band.

In 3C they present the mRNA of mAAT, please explain why this is shown and not for instance b-actin the reference gene?

The signal of the reference gene is not given. Unacceptable ! All in all the quality of these Northern blots is highly questionable. The absence of clear bands like with mAAT, indicate RNA degradation

making it not suitable for a reliable quantification.

In addition, it is not clear how many mice and how many samples were used for this and for giving the ratio's given in the table.

Cypor deficiency affects liver metabolism and to show this APAP based selection will be safe for patients additional analysis should be done. The Cypor deficient mouse shows fat accumulation in the liver. The authors perform a histological staining to investigate if this occurs, in the corrected hepatocytes. In addition, other histologic analysis needed: HE staining, staining for fibrosis.

The experiments showing the Homologous Recombination in weaker expressing genes. The methods given state adult mice 3 to 10 weeks old receive the vector. In the text the mention 6 to 8 weeks old mice. Please explain which is the correct.

In addition, please give the dose in vg/kg in the legends to allow comparison between the doses in both groups.

Finally, the timing is very different. In neonates the wait 8 to 10 weeks, in the adults just 2 to 3 weeks. This makes a comparison between both also difficult and may in fact result in under estimation of the efficacy in adults and does not allow stating that the difference is only due to no hepatocyte proliferation, which may be the case but is not due to the very different timing is not supported by these data.

Figure 4:

They should show co-expression of hFIX in the TdTomato positive hepatocytes; This should be possible at least for Alb and possibly for Ttr.

The gating differs between the 5 experimental groups. Is that really allowed in such a comparison?

They used a Mann-Whitney test to compare these five different groups. Was this somehow corrected for repeated testing? Is this the proper statistic test for these data?

Discussion:

The idea of using Cypor to select corrected hepatocytes is great but since Cypor affects all the Cyps the potential consequences should be discussed in greater detail.

Points to consider:

As mentioned above, there is fat accumulation seen in the liver specific K.O. Expanding the data showing normal/abnormal liver histology is essential to demonstrate this method will be safe. Presence of fat or other effects of absent Cyp activity, a.o. bile salt synthesis may also affect hepatocyte survival that upon stopping the APAP administration can result in loss of correction. In addition, other drugs compounds may result in toxicity specific for the corrected cells. For instance the exposure to benzo[a]pyrene of Hepatic Cytochrome P450 Reductase Null (HRN) results in more benzo[a]pyrene diol epoxide-DNA adducts (Arlt, V.M et al 2012Toxicology Letters. In addition it affects 2-amino-1-methyl-6-phenylimidazo[4,5-b]pyridine-DNA adduct formation (Arlt, V.M. et al 2011Drug Metabolism and Disposition). Furthermore, in man these systems are very heterogeneous which may render the proposed selection quite complicated.

Reviewer #4 (Remarks to the Author):

The study used scgRNA system to create drug-selectable gene editing events in specific hepatocyte loci. Below are specific comments

Major points:

1. Why does a second AAV injection in mice still work? I suppose that the second AAV has immune responses to it so it will not work. Does the first AAV injection have an effect on the second AAV injection in the manuscript?
2. What is the range of therapeutic genes that can be accommodated in the rAAV-GR vector due to the limitations of the package size of AAV(4.7k)?
3. Does hF9 play a therapeutic role in the two animal models in the manuscript? Are there any identified diseases that can be treated with this system?
4. Please explain the process of establishing a library in RNA sequencing, and the key points to evaluate the results

5. Please use deep sequencing for quantification of the data in the manuscript.
6. The animal control group in the manuscript is not rigorous
7. Flow cytometry data is suitable, and sorting and sequencing of tdTomato positive hepatocytes is recommended
8. Compared with the expression levels of different genes, the percentage of expressed genes in hepatocytes and the differences in integration caused by homology arms may also be important factors affecting the results.
9. Please supplement the homology arm sequences used in the manuscript

Minor points :

1. Please re-draw Figure 1A
2. Please supplement the picture of the control group in Figure 1F
3. Please replace the abscissa of Figure 2B to avoid misunderstanding
4. Please check carefully Figure 4H
5. Please check the legend to Supplementary Figure 3
6. Please check the description of this sentence in the manuscript.
" If the targeted locus yields sufficient amounts of functional gRNA, the tdTomato reporter will be activated in hepatocytes with homologous recombination events."
7. Please provide a detailed description of rAAV8-LSP-spCas9.

REVIEWER COMMENTS

Reviewer #1 (Remarks to the Author):

In this paper, the team of Grompe use self-cleaving guide RNA (scgRNA) using ribozyme motifs to create drug-selectable gene targeting events in hepatocytes. To achieve this, the authors deliver an AAV vector containing a promoter-less therapeutic gene flanked with homology regions that enable targeted integration in the albumin locus, co-opting this endogenous promoter for its expression. In addition, the same AAV construct also contains the scgRNA-encoding gene. Consequently, the scgRNA is also integrated into the albumin locus and is expressed from the endogenous albumin promoter. Hence, this approach allows for tissue-specific expression of gRNA obviating the need to rely on aspecific, ubiquitously expressed RNA pol III promoters (such as U6). The scgRNA was designed to target genes that when targeted confer a selective survival advantage to the gene-modified hepatocytes in the presence of a selective drug. This was validated in the FAH^{-/-} tyrosinemia mouse model after selective targeting of the Hpd gene by scgRNA and Cas9 and upon withdrawal of NTBC. An alternative model was also used based on acetaminophen-selection of hepatocytes in which the cytochrome p450 reductase (Cypor) had been inactivated by scgRNA/Cas9. Using 2 different models significantly strengthens the study.

Given the low efficiency of gene targeting by homologous recombination in hepatocytes there is an unmet need in the field to select for gene-edited cells in vivo. The current data establish proof of concept that scgRNA can be used for selective enrichment of gene-edited hepatocytes in vivo resulting in therapeutic expression levels of a gene of interest (e.g. factor IX). To my knowledge, this is the first report that exploits the scgRNA-mediated gene targeting and the use of Cypor as a targeting strategy for pharmacological acetaminophen-mediated in vivo selection of gene-edited hepatocytes with therapeutic consequences. In vivo selection of gene-edited hepatocytes in the FAH model is well established but the development of in vivo selection strategies that go beyond FAH significantly broaden the scope of applications in gene editing. Though the authors previously used the same approach to select for lentivirally-transduced hepatocytes in vivo, the current study goes significantly beyond this by demonstrating efficient in vivo selection of gene-targeted hepatocytes that are in limited abundance compared to lentivirally transduced cells. The methodology is sound and the findings are novel.

Comments

1. What are the long-term consequences and safety aspect of inactivating Cypor. This should preferably be addressed experimentally after selection of the Cypor-negative cells. Transgenic mouse studies based on selected inactivation of Cypor indicate significant increases in liver weight, accompanied by hepatic lipidosis, abnormal cholesterol homeostasis and other pathologic changes, especially long-term (after 2 months) (Gu et al., J Bioch Chem. 278:25895, 2003).

The issue of long-term safety of partial Cypor deficiency was addressed experimentally in our previous publication (Vonada et al., *Science Translational Medicine*, 13(597), 2021). Briefly, mice were APAP selected, and following complete selection mice received no further APAP treatment for 42 weeks before harvest. A complete liver and lipid panel performed on blood taken at harvest showed no significant differences between these animals and naïve controls, despite the presence of ~ 30% Cypor null hepatocytes. This analysis includes plasma cholesterol and triglycerides. These experimental mice showed no loss of Cypor-deficient hepatocytes as compared to animals that received continued weekly APAP treatment over the 42-week period, indicating no disadvantage for the Cypor-deficient cells in the absence of APAP. Although mice did show mild accumulation of lipids in Cypor-deficient areas, liver histology revealed no signs of hepatocellular injury, inflammation, or necrosis.

A notable difference from the mice described by Gu et al. (*J Bioch Chem.*, 278:25895, 2003) is that Gu et al. described animals with a Cypor knockout in all hepatocytes, whereas Cypor deficiency created by APAP selection is limited to a subset of hepatocytes in zone 3 of the hepatic lobule. This limitation is due to zone 3 specific expression patterns of the specific Cyp enzymes (Cyp2E1, Cyp1A2) that are required for the metabolism of APAP to NAPQI. This limitation to only zone 3 is a safety feature of our system as it is not possible for APAP selection to lead to a loss of Cypor in all hepatocytes. This may also explain why our mice did not show altered plasma cholesterol and triglycerides as described by Gu et al.

2. I appreciate that hemophilia is used as a model to establish proof of concept. However, since long-term efficacy and safety has been established following conventional AAV-based gene therapy (not gene editing) in severe hemophilia B patients in clinical trials, the case for using this approach for this particular disease is not very convincing. Nevertheless, the targeting and selection approach may have merit in pediatric subjects to overcome the limitation of diluted AAV genomes upon hepatocyte cell division. However, the substantial cell death that occurs following administration of toxic doses acetoaminophen is a concern in disease like hemophilia for which there are other treatment options and given the advances in conventional gene therapy. This strategy would ultimately be better suited for severe lethal diseases of high unmet medical need for which there are no effective therapies available. This should at least be discussed with a clear focus on which diseases may ultimately benefit from this strategy and for which the hepatotoxicity following in vivo selection would be considered acceptable.

Our approach is novel and therefore we chose the most convenient system to demonstrate proof-of-principle. The ability to track human F9 noninvasively was the main reason for choosing this molecule. We agree that other genetic liver diseases may have a more compelling need and have given our reasons for choosing human F9 for the experiments in our current manuscript. Please also see the response to reviewer 4, Major Point #2.

3. The authors employ a promoter-less construct to achieve targeting expression of the therapeutic gene based on prior work previously published by M. Kay's group (Barzel et al. *Nature* 517: 360-4, 2015). The merit of that system (designed as "GR" or "gene-ride" in the manuscript) is that it enables gene targeting in the absence of CRISPR (or any other nuclease-based strategy to achieve targeted double strand DNA breaks) to circumvent possible

limitations of using CRISPR for gene targeting (i.e. off-target effects, immune clearance of gene-targeted cells due to the immunogenicity of Cas9 etc.). Hence, the CRISPR Cas9-independence is the main attribute and “sales-pitch” of this GR platform technology. However, in the current study, CRISPR Cas9 was introduced into the hepatocytes to achieve targeted scgRNA-mediated disruption of the Hpd or Cypor genes in order to confer a selective advantage to the gene-edited cells. Consequently, the main advantage of using the GR platform in the first place is eclipsed by the reintroduction of the CRISPR system into the target cells. In this case it is not clear what the main advantage is of relying on the GR platform, given the low overall efficiency of CRISPR-independent targeting in the albumin locus. Since it is expected that the targeting efficiency of the promoter-less construct can be increased using CRISPR/Cas9 and since Cas9 is already present in the hepatocytes, it would make more sense to assess the potential for multiplexing using scgRNA that are specifically designed to target: (i) the Hpd or Cypor genes to enable selection of gene-edited hepatocytes and (ii) the albumin locus to enhance the efficiency of targeted integration of the FIX gene downstream of the endogenous albumin promoter.

The reviewer is correct that our approach requires spCas9, at least transiently. However, the efficiency of gene targeting still remains subtherapeutic (5% or less) in adult liver after induction of DSB the target (Barzel et.al., *Nature*, 517:360-4, 2015). Hence, APAP selection is still advantageous.

In the GeneRide vector, the scgRNA can only be expressed after integration into the target locus and therefore, inclusion of a scgRNA for the target locus (Albumin) into our GR-AAV would not enhance homologous recombination. The Albumin gRNA could be included in the spCas9 AAV, but we believe that the clinical application of our method would not use a spCas9-AAV, as employed herein for our proof-of-principle experiments. We envision transient delivery of spCas9 RNA with a nanoparticle.

One of the main benefits of the GR vector is that it is promoterless and this feature is maintained in GR scgRNA system. The promoterless nature of the GR vector is beneficial as it can protect against undesired gene expression and potential activation of oncogenes in the event of a random integration. In the current GR scgRNA system we describe, only one double strand break is made as result of the Cypor guide RNA. Although a low frequency of initial targeted integration events are achieved, the selection process allows more than sufficient expansion of the hepatocyte population with the desired homologous recombination event. It can be detrimental to create 2 double strand breaks, as double strand breaks in 2 or more loci can result in translocation or rearrangements of the chromosomal DNA (Shin et. al., *Nature Communications*, 8:15464, 2017). Additionally, the system described here would be amenable to base editing that would allow loss of Cypor without the creation of any double strand breaks using a base editor such as CRISPR-X or activation induced deaminase-mediated mutagenesis to create multiple modifications to Cypor in vivo (Rees & Liu, *Nature Reviews Genetics*, 19: 770-788, 2018).

4. The authors claim that one of the main advantages of the platform is that it enables expression of the gRNA in a tissue-specific manner by co-opting the endogenous RNA polymerase II albumin promoter upon targeted integration as opposed to using ubiquitously expressed RNA pol III promoters to drive the gRNA expression (like U6 or H1). The concept of

using pol II promoter to drive sgRNA has been published previously and prior publications were cited by the authors. In addition, the sgRNA concept has been used for targeted upregulation of therapeutically relevant proteins (e.g. Xu et al. *Nucleic Acids Res* 2017 Mar 17;45(5):e28). The advantage of expressing the sgRNA in a cell type-specific manner in vivo to achieve cell type specific targeting is not a gamechanger since this has been made possible by expressing the Cas9 protein itself from a tissue-specific promoter, particularly in the liver. What is the promoter used in the current studies to drive Cas9 in the different models? This was not specified.

The main advantage of scgRNA as compared to Cas9 itself is size. Our technology aims to make precise gene editing events created by homologous recombination selectable and rAAV vectors are currently the best homology donor for homologous recombination in hepatocytes in vivo. rAAVs have a packaging limit of 4.7kb and spCas9 is much too big to fit into rAAVs along with the therapeutic transgene and homology arms. While our current work uses the tiny liver specific promoter (Chuah et al, *Molecular Therapy*, 22(9): 1605-1613, 2014) to drive the expression of spCas9 in a rAAV vector, the use of a cell type specific scgRNA would allow the use of other strategies to deliver spCas9 transiently. These include nanoparticles, nanoclews or ultrasound mediated bubbles. The transient presence of spCas9 is preferred as it can prevent off target nuclease activity or immunogenicity that can result from long term continuous expression of spCas9.

5. Fig S3: Though TTR is a weaker promoter than albumin it is still considered a very strong hepatocyte-specific promoter that has therefore been used in gene therapy clinical trials. The case for demonstrating the potential of this technology in the context of weaker promoter is not very strong, particularly since the effectiveness of the strategy has not been demonstrated when the FIX endogenous promoter was targeted.

We wished to explore the dynamic range of our methodology, because more moderate levels of gene expression may be preferable for some therapeutic transgenes. We agree with the reviewer that the case for exploring weaker promoters is not strong for secreted protein transgenes. However, physiologic levels of transgene expression may be critical for the success of gene therapy in other diseases with cell-autonomous pathophysiology. Superphysiologic overexpression of a therapeutic transgene can be toxic, depending on the disease. Our experiments provide proof-of-principle that targeting other loci is achievable with our method and could be applied depending on the target disorder.

6. It would appear that the FIX levels attained without selection are much lower than what has been published previously (Barzel et al. *Nature* 517: 360-4, 2015). What is the reason for this difference?

The vector that we initially used for our experiments was from the original Barzel et al. plasmid with the addition of the scgRNA. In our initial experiments we found similar hF9 baseline results as reported by Barzel. However, a publication by Ian Alexander's group in 2017 (Logan et al., *Nature Genetics* 49:1267-1273, 2017) demonstrated that the presence of a cryptic promoter in rAAV2 based constructs could drive expression of the transgene present. Upon checking the vector obtained from Barzel, the sequence for the cryptic promoter was found to be present in the 3' region. We removed the cryptic promoter sequence by subcloning the region between

the ITRs into a rAAV backbone obtained from Cell BioLabs that was confirmed to be cryptic promoter free. We have changed the methods to include this. Following the removal of the ITR we observed lower baseline hF9 levels suggesting that the initially higher baseline levels of hF9 observed may be due to the presence of the cryptic promoter driving expression of rAAV concatemers in the original Barzel vector.

7. Do Cypor mRNA levels correlate with the targeting efficacy?

Cypor mRNA levels were not determined as only 10-20% of Cypor indel in hepatocytes from each animal were observed, hence we would expect an approximately 10-20% decrease in gene expression. This slight difference is too small for detection by qRT-PCR.

8. Is the gene inactivation of the Cypor and Hpd genes mono-allelic or bi-allelic? What about the targeting at the albumin locus ?

Gene inactivation of Cypor and Hpd should be completely biallelic. When performing immunofluorescent staining for Cypor and hF9, we see clonal expansion where the hepatocytes are Cypor negative and hF9 positive. If there were monoallelic gene inactivation of Cypor, we would expect to observe clones that are hF9 positive and Cypor positive.

9. It is difficult to reconcile the high levels of FIX protein with the very low FIX mRNA levels. This does not quite add up. It would be interesting to ascertain in a controlled experiment whether the low FIX mRNA levels are due to the incorporation of the scgRNA cassette

We used two different RNA quantification methods and are confident in our results. Based on previous RNA sequencing of wild type mice performed, we have observed that endogenous mRNA Factor 9 levels in a mouse are approximately 400 fold lower than Albumin. While, mRNA expression of hF9 driven from the targeted Albumin promoter is approximately 10-fold lower than Albumin, it is still 40-fold higher than the endogenous locus.

10. The authors ascribe fact that the ribozymes cannot be detected due to their instability but there is no evidence to support this. It cannot be ruled out that this is due to a technical issue

We agree with your point and have changed our discussion accordingly.

11. In the targeting construct, why is there a central region of homology to the target sequence? Does it suffice to have homology regions at the 5' and 3' end only?

The vector proven to be effective by Barzel et al. (Nature 517: 360-4, 2015) contained this central region of homology, and thus we retained this homology region for our experiments. Although we expect that the 3' and 5' regions of homology would be sufficient for targeted integration we have not directly tested this vector without the central region.

12. The tdTomato mouse constitutively express Cas9. It can be expected that the targeting efficiency would be much lower if clinically relevant AAV-Cas9 vector doses would be administered instead of using a Cas9 transgenic model.

The experiments here give insight into the maximal integration potential using this strategy. In a clinical setting other methods such as lipid nanoparticles would be the method of choice for delivering spCas9.

Reviewer #2 (Remarks to the Author):

Tiyaboonchai and colleagues presents the continuation of previous work aimed at pharmacologically selecting hepatocytes genetically modified with the GeneRide platform. In previous work, the authors used RNA mediated selecting to enable drug selection of modified hepatocytes. Here, they use the co-expression of guide RNA to enable CRISPR mediated drug resistance. The work is interesting as it could help improve levels of transgene expression obtained with the GeneRide system.

Major points:

- Give the previous work published by the authors, it is important to clearly outline the points of novelty of the current work. The basic mechanism exploited to allow for hepatocyte selection is in fact the same. One point to carefully address is that the method presented here has potential drawbacks, including the fact that a double strand cut in the genome can lead to genome instability, particularly in a setting of induced toxicity and cell proliferation, and that in large animal models and human the starting efficiency of AAV transduction in liver is remarkably smaller than in mice, thus making the approach more challenging (e.g., requiring prolonged administration of hepatotoxic drugs to achieve therapeutic levels of expression).

We have made edits in the manuscript, highlighting the novelty aspects as well as drawbacks of our method. The other reviewers specifically commented on the fact that our approach is novel. Although our APAP method was reported previously, selective expansion of precisely edited hepatocytes using self-cleaving guides is clearly novel. In fact, we have filed a patent application on the technology. Many papers have sought to develop methods to enhance the poor efficiency of homologous recombination in vivo, mostly using small molecule inhibitors of DNA repair processes. Our approach is novel and also unique. There are no other publications on using drug selection to expand gene edited cells. Our current study focuses on the use of a scgRNA in place of a shRNA. Guide RNAs can be beneficial as they result in complete loss of function of the target gene and multiple algorithms are available that can aid in efficient designs. Also, unlike shRNAs, gRNAs do not require a strong promoter for their expression. These aspects are highlighted in the discussion.

We also acknowledge the theoretical down-sides of our method, including potential genotoxicity. However, in both our current study with gRNAs and an SpCas9 nuclease and in the previous study with shRNAs, no tumors were observed in any of the mice (n>50) that had a loss of Cypor followed by APAP selection. In future work, the use of a transiently expressing spCas9 such as delivery via lipid nanoparticle, could decrease the chances of prolonged genome instability. This is in line with published literature by numerous other labs using Cas9 in vivo, including in non-human primates. No liver tumors have been reported. In addition, there is no increased risk for liver cancer in human patients that have experienced acetaminophen toxicity.

It should be noted that cell expansion and selection using moderate toxicity regimens is the mainstay of bone marrow transplantation and hematopoietic gene therapy. Without conditioning regimens the therapeutic cells remain too rare to have an effect. Hence, clinical medicine has already embraced this approach. In fact, one could argue that moderate APAP toxicity is a lot less invasive than total body irradiation and chemotherapy.

- Here two AAV vectors are administered, one containing the promoterless targeting sequence and the scgRNA and the other the Cas9. What is interesting is that the second vector still appears to work, even in the potential presence of antibodies induced by the first vector. Complete characterization of antibody titers and Cas9 expression (compared to naïve mice) is warranted. Also, the authors should point out that the approach is hardly translatable and will require the use of LNPs or other systems.

Thank you for your comments. Yes, our current work is a proof-of-principle paper and the eventual translation into the clinic will require nanoparticles instead of a spCas9 AAV. In our prior work on gene repair in tyrosinemia (Paulk et al., *Hepatology* 51:1200-1208, 2010 and Zhang et al., *Human Gene Therapy* 32:294-301, 2021), we initially used AAV as method of delivery for spCas9 but subsequent work has switched to nanoparticles for the delivery of spCas9 (Yin et al., *Nature Biotechnology* 34:328-333, 2016). Now that we know this approach works in principle, we will use more translational versions of the technology in the future such as lipid nanoparticles, DNA nanoclews (Sun et al., *Science Advances* 6(21), 2020), or ultrasound mediated gene delivery of microbubbles (Manson et al, *Blood Advances* 6:3557-3568, 2022). Transgene immunity and antibody levels in mice are not likely relevant or predictive for the potential clinical use of the methodology. We do not think that would provide critical information. The first AAV injection of the promoterless targeting sequencing used AAV serotype DJ and the second AAV with the spCas9 used serotype 8. We utilized different serotypes at each time point because we also anticipated that the first vector may introduce neutralizing antibodies preventing the second vector from working. Serotype information has been added to the methods section.

- Figs 1 and 2. Given the high vector dose used, it would be important to characterize the integration pattern of the GeneRide vector and albumin protein levels. They should also quantify immunohistochemistry images to understand what is driving FIX expression and make sure all readouts correlated.

Precise integration of the GeneRide vector was confirmed by PCR covering part of the integrated vector and the homology arm. One primer was designed to be located in the p2A inside the targeting construct and the other primer was located outside of the arm of homology. Immunofluorescence staining for hFIX was quantified for both the mice that received the GR-Hpd scgRNA ($74.9\% \pm 8.0\%$) and the GR-Cypor scgRNA ($17.2\% \pm 3.9\%$). This correlates with the TIDE results for indel frequency in Hpd and Cypor respectively that were reported in the manuscript. This data has been added to the results section and is in Table S1. Western blots were performed against Albumin protein and relative expression to Gapdh was analyzed. This data has been added to supplemental figure 3A and 3B. We found no significant difference in Albumin at a protein level in animals that received the GR vector and did or did not undergo selection whether by NTBC withdrawal in the case of the Hpd-scgRNA or APAP treatment in the Cypor-scgRNA cohort. This corresponds to the results from next generation RNA sequencing also demonstrating no difference in the Albumin expression in selected or unselected mice.

- Figure 4 is confusing, seems like results do not correlate with the expected expression

strength of the genes selected, as for example TTR is better than ALB in this model. Were the mice treated with acetaminophen? If not, it would be important to administer acetaminophen to compare results with Figs 1 and 2 and particularly to compare what happens in the GeneRide vs. no homology groups. The levels of targeting seem also low, are they like those in Fig 1-2? The actual dose of vector was quite similar (8E10 vs. 1E11 vg). As discussed above, antibodies to AAV should be measured in all animals as well as Cas9 expression. For the FIX gene, low expression of the scgRNA and low expression of Cas9 could explain the background levels of TdTomato+ cells.

The experiment in figure 4 does not include acetaminophen selection, because the vector was designed to activate a reporter gene, not inactivate Cypor. We have made this clearer in the figure legend of our revision and text of the results section. In this experiment the mice received the GR vector containing arms of homology to the various genes (Alb, Ttr, Pah, F9 and no homology). Within the arms of homology is a self-cleaving guide RNA containing gRNA targeting the stop cassette upstream of tdTomato in the Ai9-tdTomato mouse, and not against Cypor. The purpose of the experiment was to determine if a scgRNA could function from a lower expressing endogenous promoter. The results of the experiment as you mentioned, do not correlate with what would be expected if there was a linear correlation of scgRNA expression level and CRISPR/spCas9 nuclease activity. Instead, the result shows that the dynamic range of scgRNA is very large and that even tiny amounts of gRNA are sufficient for the enzyme to work. This is a very favorable property of our system: it is not very dose sensitive. The differences in reporter gene activation seen with the different genes is likely due to the integration frequency itself. It is unsurprising that recombination frequencies at different chromosomal loci are different and not necessarily correlated with the transcriptional strength of the target locus. Our explanation in the text of the results and the discussion section has been adjusted for clarity and to include these additional possibilities.

Reviewer #3 (Remarks to the Author):

The authors report a method that allows to provide a growth advantage to hepatocytes upon correct insertion of a therapeutic gene, FIX, by gene ride. Upon selection the efficacy of the approach is significantly increased providing therapeutic correction. The approach taken follows up on previous studies of this group.

Comments:

The data presented in fig 1 and 2 are straightforward showing efficacy of the strategy.

I do have major problems with figure 3 that need to be clarified and improved.

In 3 A the signal for mAlb is shown. The levels are very different and strangely, no signal is detected in the non-selected animals. Please explain. The quantification given in the table does present very different results and only a limited spread in expression. How was this calculated.

We apologize for the confusion. In Panel 3A we are using a probe against mouse Albumin. These samples have a slightly unequal load but the main conclusions we wanted to draw from this blot is that a larger band representing the mRNA of Albumin fused to the hFIX transgene

and the scgRNA cannot be seen at this exposure level. The Albumin-hF9 fusion mRNA should be visible if the targeted allele was expressed as highly as the untargeted locus.

To clarify, the quantification on the table in Figure 3D is not quantification of the Northern Blot expression, but instead uses next generation RNA sequencing because it is more quantitative. We will add some titles and edit the figure legend to figure 3 for clarity.

In 3B although only in the selected animals the mRNA with the correct size is found, as expected, again the levels are completely different from no signal to a clear band.

In figure 3B we are measuring human F9. Human F9 is not expressed endogenously by mouse tissue and the probe we are using in the northern blot is specific to human F9. In the absence of APAP selection (the unselected group), there should be very few hepatocytes with the integrated GR vector containing human F9 and the Cypor scgRNA. We expect that these human F9 levels would be too low to detect by northern blot and that is the reason for a lack of signal in the unselected group of mice.

In 3C they present the mRNA of mAAT, please explain why this is shown and not for instance b-actin the reference gene?

Since we are analyzing gene expression in hepatocytes, mouse AAT was chosen as it is a liver specific gene that is uniformly expressed in all hepatocytes and not in other cells in the liver such as the nonparenchymal cells. This probe was chosen to demonstrate that RNA quality is sufficient for northern blotting across the different samples.

The signal of the reference gene is not given. Unacceptable ! All in all the quality of these Northern blots is highly questionable. The absence of clear bands like with mAAT, indicate RNA degradation making it not suitable for a reliable quantification.

RNA quality was checked by clear visualization of 28S, 18S and 5S rDNA bands on an ethidium bromide stained agarose gel (below), as is typical for northern blots. The RNA was not degraded.

However, rather than solely relying on the northern blots for quantification of the mRNA, we used the northern blots as a qualitative measure of the presence of these mRNAs as northern blots demonstrate the size of the mRNA. To be able to reliably quantify the amount of mRNA expression we decided to use next generation RNA sequencing. RNA sequencing can result in more accurate quantification of the mRNA and the results from those experiments are shown in panel 3D.

In addition, it is not clear how many mice and how many samples were used for this and for giving the ratio's given in the table.

In the northern blots, each lane represents sample from 1 mouse. For the RNA sequencing in Panel 3D, three unselected mice and three APAP selected mice were used for the next generation RNA sequencing. This information has been added to the results section, figure legend and methods section.

Cypor deficiency affects liver metabolism and to show this APAP based selection will be safe for patients additional analysis should be done. The Cypor deficient mouse shows fat accumulation in the liver. The authors perform a histological staining to investigate if this occurs, in the corrected hepatocytes. In addition, other histologic analysis needed: HE staining, staining for fibrosis.

This is an important comment. Thank you. See response to reviewer 1, comment #1 as well. The issue of long-term safety of partial Cypor deficiency was thoroughly examined in our previous publication (Vonada et al., Science Translational Medicine, 13(597), 2021). Briefly, mice were APAP selected, and following complete selection mice received no further APAP treatment for 42 weeks before harvest. A complete liver and lipid panel performed on blood taken at harvest showed no significant differences between these animals and naïve controls. This includes plasma cholesterol and triglycerides. These experimental mice showed no loss of Cypor-deficient hepatocytes as compared to animals that received continued weekly APAP treatment over the 42-week period, indicating no disadvantage for the Cypor-deficient cells in the absence of APAP. Although mice did show mild accumulation of lipids in Cypor-deficient areas, liver histology revealed no signs of hepatocellular injury, inflammation, or necrosis. The lipid accumulation phenotype was much less severe than that described by others in mice with complete hepatic Cypor deficiency (Gu et al 2003, Henderson et al 2003). This difference may be due to the fact that our selection system is limited to hepatocytes in zone 3 of the hepatic lobule (less than half of all hepatocytes) due to zone 3-specific expression of APAP-metabolizing Cyp enzymes. This is a safety feature of our system as it is not possible for APAP selection to lead to a loss of Cypor in all hepatocytes.

H&E staining was performed and in the experimental mice compared to the controls that did not receive APAP there mild accumulation of lipids is seen. Fibrosis was assessed by staining liver tissue sections with picro Sirius red. Mild fibrosis was observed in the mice that received the Cypor-scgRNA GR rAAV and APAP selection compared to the mice that did not receive any APAP or were wild type naïve mice. Images from the H&E and picro Sirius red staining have been added in the figures.

The experiments showing the Homologous Recombination in weaker expressing genes. The methods given state adult mice 3 to 10 weeks old receive the vector. In the text the mention 6 to 8 weeks old mice. Please explain which is the correct.

Thank you for catching this. The adult mice were 3 to 10 weeks old when receiving the vector. This has been corrected in the text.

In addition, please give the dose in vg/kg in the legends to allow comparison between the doses in both groups.

This has been added to the figure legends and methods section

Finally, the timing is very different. In neonates the wait 8 to 10 weeks, in the adults just 2 to 3 weeks. This makes a comparison between both also difficult and may in fact result in under estimation of the efficacy in adults and does not allow stating that the difference is only due to no hepatocyte proliferation, which may be the case but is not due to the very different timing is not supported by these data.

In these experiments, we are not comparing the integration of the AAV in different age groups. Rather we are comparing the effect of each GR AAV integrating into different loci based on the different homology arms on the expression of the scgRNA.

Figure 4:

They should show co-expression of hFIX in the TdTomato positive hepatocytes; This should be possible at least for Alb and possibly for Ttr.

We did show co-expression of hFIX and the expression of the scgRNA in Figure 2. The hFIX cDNA was included in our constructs for the experiment in Figure4 only to make sure that the vectors differed only by their homology arms. We did not intend to measure transgene expression, because the weaker promoters are unlikely to give enough signal for Immunohistochemistry.

The gating differs between the 5 experimental groups. Is that really allowed in such a comparison?

Not all mice (>100 animals) were perfused for single cell isolation and analyzed by flow cytometry on the same day. Gates differ slightly from day to day based on controls run on that day.

They used a Mann-Whitney test to compare these five different groups. Was this somehow corrected for repeated testing? Is this the proper statistic test for these data?

Thank you for catching this. The correct statistical test is a one way ANOVA with Tukey's multiple comparison test. This statistical test has been performed and the data has been updated in the figure as well as added to the methods.

Discussion:

The idea of using Cypor to select corrected hepatocytes is great but since Cypor affects all the Cyps the potential consequences should be discussed in greater detail.

Points to consider:

As mentioned above, there is fat accumulation seen in the liver specific K.O. Expanding the data showing normal/abnormal liver histology is essential to demonstrate this method will be safe. Presence of fat or other effects of absent Cyp activity, a.o. bile salt synthesis may also affect hepatocyte survival that upon stopping the APAP administration can result in loss of correction. In addition, other drugs compounds may result in toxicity specific for the corrected cells. For instance the exposure to benzo[a]pyrene of Hepatic Cytochrome P450 Reductase Null (HRN) results in more benzo[a]pyrene diol epoxide-DNA adducts (Arlt, V.M et al 2012Toxicology Letters. In addition it affects 2-amino-1-methyl-6-phenylimidazo[4,5-b]pyridine-DNA adduct

formation (Arlt, V.M. et al 2011 Drug Metabolism and Disposition). Furthermore, in man these systems are very heterogeneous which may render the proposed selection quite complicated. Please also see response to reviewer 1, comment #1. We have previously addressed whether Cypor deficiency will confer a selective disadvantage in the absence of acetaminophen (Vonada et al., Science Translational Medicine, 13(597), 2021). Mice were APAP selected, and following complete selection mice received no further APAP treatment for 42 weeks prior to harvest. These experimental mice showed no loss of Cypor-deficient hepatocytes as compared to animals that received continued weekly APAP treatment over the 42-week period, indicating no disadvantage for the Cypor-deficient cells in the absence of APAP. Additionally, a complete blood lipid and liver panel showed no differences between these animals and untreated controls, indicating no adverse health effects of partial Cypor deficiency. Human patients with Cypor deficiency do not show a liver phenotype. This indicates that partial Cypor deficiency is likely to be well tolerated.

Reviewer #4 (Remarks to the Author):

The study used scgRNA system to create drug-selectable gene editing events in specific hepatocyte loci. Below are specific comments

Major points:

1. Why does a second AAV injection in mice still work? I suppose that the second AAV has immune responses to it so it will not work. Does the first AAV injection have an effect on the second AAV injection in the manuscript?

The first AAV injection used AAV serotype DJ and the second used serotype 8. This information has been added to the methods under the rAAV production and delivery section.

2. What is the range of therapeutic genes that can be accommodated in the rAAV-GR vector due to the limitations of the package size of AAV(4.7k)?

Assuming 1 kb each as the standard size for homology arms and given the ~300 bp size of the scgRNA, about 2.4 kb of space for therapeutic payload remain. Hence proteins of up to 800 amino acids in size could be accommodated. Some examples include for Branched chain ketoacid dehydrogenase E1, alpha polypeptide (446 amino acids) or Branched chain ketoacid dehydrogenase E1, beta polypeptide (390 amino acids) for maple syrup urine disease, Alpha-galactosidase A (size 421 amino acids) for Fabry disease, Phenylalanine hydroxylase (452 amino acids) for Phenylketouria, Homogentisate oxidase (445 amino acids) for Alkaptouria and Glucocerebrosidase (536 amino acids) for Gaucher disease.

3. Does hF9 play a therapeutic role in the two animal models in the manuscript? Are there any identified diseases that can be treated with this system?

Factor 9 is the protein deficient in patients with hemophilia B and the system described herein could be used to treat this disease. However, the animals used here do not have a deficiency in murine F9 and we used hF9 protein levels rather than functional disease correction as a metric for achieving therapeutic threshold. We demonstrate that selection of hepatocytes with the

integrated hF9 can result in expansion of the small initial population to therapeutic levels if the animal did have a deficiency in F9. Furthermore, due to the circulatory nature of F9, it is also a good biomarker for tracking progress of the selection process. While this approach could be therapeutic for hemophilia B, other genetic liver diseases with greater unmet medical need are more compelling targets. Discussion of these potential target diseases has been added to the paper.

4. Please explain the process of establishing a library in RNA sequencing, and the key points to evaluate the results

This has been updated in the methods under the RNA sequencing and data analysis section.

5. Please use deep sequencing for quantification of the data in the manuscript.

We do use some deep sequencing for quantification of the data in Figure 3D.

6. The animal control group in the manuscript is not rigorous

Immunofluorescent staining for hF9 in the mice that received the GR containing the scgRNA but did not receive spCas9 has been added to the revised paper.

In figure 2, data from additional control mice that received only the GR-Cypor scgRNA (without any spCas9) or saline that then received acetaminophen treatment were not shown as high doses of acetaminophen are hepatotoxic and the majority of the mice in these groups were found either deceased or had to be euthanized at different time points throughout the experiment. This data has been added as supplemental figure 2.

7. Flow cytometry data is suitable, and sorting and sequencing of tdTomato positive hepatocytes is recommended

To address whether homologous recombination of the targeting vectors has occurred, PCRs which confirm integration into their targeted loci have been performed and are included in supplemental figure 5.

8. Compared with the expression levels of different genes, the percentage of expressed genes in hepatocytes and the differences in integration caused by homology arms may also be important factors affecting the results.

This is a valid point and we agree that the one factor affecting difference in integration could be the differences in homology arms themselves. We have added this to the discussion.

9. Please supplement the homology arm sequences used in the manuscript

The sequence of the homology arms is now indicated in the methods.

Minor points :

1. Please re-draw Figure 1A

Figure 1A has been re-drawn.

2. Please supplement the picture of the control group in Figure 1F

An IF image with hF9 staining from the control (w/o Cas9) group has been added to Figure 1F.

3. Please replace the abscissa of Figure 2B to avoid misunderstanding

The graph has been redrawn with adjustments to the abscissa and replaced in Figure 2B.

4. Please check carefully Figure 4H

A one way ANOVA with Tukey's multiple comparison test instead of a regular t-test has been used for data in this figure. The data has been updated in the figure as well as added to the methods.

5. Please check the legend to Supplementary Figure 3

The legend of Supplementary figure 3 has been fixed.

6. Please check the description of this sentence in the manuscript.

" If the targeted locus yields sufficient amounts of functional gRNA, the tdTomato reporter will be activated in hepatocytes with homologous recombination events."

This sentence has been edited in the manuscript as follows: Unlike in the drug selection experiments described above, homologous recombination into the target locus of and expression of a functional gRNA will result in tdTomato reporter activation, but not APAP resistance.

7. Please provide a detailed description of rAAV8-LSP-spCas9.

The rAAV8-LSP-spCas9 vector consists of the tiny liver specific promoter (LSP) [Chuah et al., 2014] (a gift from Mark Kay) driving the expression of spCas9. spCas9 was restriction digested from the pX330-U6-Chimeric_BB-Cbh-hSpCas9 plasmid (a gift from Feng Zhang, Addgene plasmid #42230; <http://n2t.net/addgene:32230>; RRID: Addgene_42230) using AgeI and EcoRI and ligated into an AAV vector plasmid. The LSP promoter was PCR amplified and then ligated into the NotI site on this plasmid. rAAV serotype 8 was produced from this final plasmid. This description of the construction of this plasmid has been added to the methods section.

REVIEWER COMMENTS

Reviewer #1 (Remarks to the Author):

The authors have adequately addressed my comments

Reviewer #2 (Remarks to the Author):

The authors addressed all concerns, no additional comments.

Reviewer #3 (Remarks to the Author):

The authors have addressed my remarks. I have no additional comments.

Reviewer #4 (Remarks to the Author):

No more comments